# Intercellular transfer of cancer cell invasiveness via endosome-mediated protease shedding

Eva Maria Wenzel [1,2], Nina Marie Pedersen[1,2], Liv Anker Elfmark[1,2], Ling Wang[1,2], Ingrid Kjos [1,2], Espen Stang[3], Lene Malerød[1,2], Andreas Brech[1,2,4], Harald Stenmark [1,2] & Camilla Raiborg [1,2] ✉

Overexpression of the transmembrane matrix metalloproteinase MT1-MMP/MMP14 promotes cancer cell invasion. Here we show that MT1-MMP-positive cancer cells turn MT1-MMP-negative cells invasive by transferring a soluble catalytic ectodomain of MT1-MMP. Surprisingly, this effect depends on the presence of TKS4 and TKS5 in the donor cell, adaptor proteins previously implicated in invadopodia formation. In endosomes of the donor cell, TKS4/5 promote ADAM-mediated cleavage of MT1-MMP by bridging the two proteases, and cleavage is stimulated by the low intraluminal pH of endosomes. The bridging depends on the PX domains of TKS4/5, which coincidently interact with the cytosolic tail of MT1-MMP and endosomal phosphatidylinositol 3-phosphate. MT1-MMP recruits TKS4/5 into multivesicular endosomes for their subsequent co-secretion in extracellular vesicles, together with the enzymatically active ectodomain. The shed ectodomain converts non-invasive recipient cells into an invasive phenotype. Thus, TKS4/5 promote intercellular transfer of cancer cell invasiveness by facilitating ADAM-mediated shedding of MT1-MMP in acidic endosomes.

A prerequisite for the expansion and subsequent metastasis of a tumour is the degradation of the surrounding extracellular matrix (ECM). Matrix metalloproteases (MMPs) are soluble or membrane-bound enzymes that digest ECM fibres, allowing tumour cells to breach the basement membrane to invade the surrounding tissues. A key component for cancer progression is the membrane-type 1 matrix metalloproteinase (MT1-MMP/MMP14), whose overexpression is associated with poor prognosis for various cancer types[1]. MT1-MMP is considered pro-invasive and pro-tumourigenic, making it an attractive target for cancer therapy[2]. It displays enzymatic activity towards type-I, -II and −III collagen, gelatin, fibronectin, laminins 1 and 5 and vitronectin, and also activates other soluble MMPs, such as proMMP2 and proMMP13[3,4], thereby contributing to ECM disintegration in a direct and indirect fashion.

The best described function of MT1-MMP is at invadopodia and podosomes, where MT1-MMP is deposited in a targeted fashion through tubulation from late endosomes[5,6]. Podosomes are small actin-rich cellular protrusions enriched for matrix metalloproteases, which digest ECM and which are a prerequisite for cell invasion of, for example, macrophages and osteoclasts. Tumour cells can develop corresponding structures, called invadopodia, which enable the cells to invade and metastasize (reviewed in[7]). For the formation of functional podosomes and invadopodia, two scaffolding proteins, SH3 and PX domain-containing protein 2A and 2B (TKS5 and TKS4), are

[1]Centre for Cancer Cell Reprogramming, Faculty of Medicine, University of Oslo, Oslo, Norway. [2]Department of Molecular Cell Biology, Institute for Cancer Research, Oslo University Hospital, Oslo, Norway. [3]Laboratory for Molecular and Cellular Cancer Research, Department of Pathology, Oslo University Hospital, Oslo, Norway. [4]Section for Physiology and Cell Biology, Dept. of Biosciences, Faculty of Mathematics and Natural Sciences, University of Oslo, Oslo, Norway. ✉e-mail: camilrai@medisin.uio.no

essential. Following growth factor signalling, TKS4 and TKS5 are phosphorylated and activated, which enables them to recognize the phospholipid phosphatidylinositol 3,4-bisphosphate (PtdIns(3,4)P$_2$) at the invadopodial membrane by their amino-terminal Phox homology (PX) domain. With their four or five SH3 domains, respectively, TKS4 and TKS5 act as adaptor proteins with numerous binding partners, including actin regulators, cortactin and ADAM (a disintegrin and metalloprotease) sheddases[8–10]. Similar to MT1-MMP, overexpression of both TKS proteins has been implicated in progression of cancers such as brain tumours, lung adenocarcinomas, prostate cancer, breast cancer and melanoma[11–15].

The transmembrane protein MT1-MMP localises to various cellular membranes and traffics from the plasma membrane through the endocytic pathway[16]. The short C-terminal tail serves as a hub for multiple cytosolic proteins and regulates MT1-MMP's function[17]. Interestingly, MT1-MMP has also been found associated with extracellular vesicles (EVs), where it is enzymatically active and can contribute to premetastatic niche formation[18–21]. In addition, MT1-MMP can undergo cleavage of its catalytically active extracellular domain, mediated by the ADAM family of sheddases[22,23]. It is not known how the ADAM-mediated processing of MT1-MMP is spatially and mechanistically controlled. The function and regulation of the soluble or EV-associated MT1-MMP is poorly understood.

Here, we find that TKS4/5 facilitate ADAM-mediated cleavage of MT1-MMP in acidic endosomes by bridging the two proteases,

thereby providing a spatiotemporal control over their activities. The subsequent secretion of the soluble and EV-associated MT1-MMP from highly invasive cancer cells turns MT1-MMP-negative cells degradative and invasive. Thus, overexpression of TKS4/5 and MT1-MMP promotes intercellular transfer of cancer cell invasiveness.

## Results

### TKS4 and TKS5 are recruited to endosomes by the cytosolic tail of MT1-MMP

MDA-MB-231 cells are highly invasive triple negative breast cancer cells, known to form invadopodia[24]. When MDA-MB-231 cells were grown on polymerized type I collagen, they formed linear invadopodia positive for mCherry-tagged MT1-MMP (mCh-MT1-MMP) and the invadopodia marker protein TKS5-GFP, as expected (Fig. 1a)[25]. In addition, both proteins were found to colocalise in cytoplasmic structures resembling endosomes (Fig. 1a). Whereas endogenous TKS5 was barely detectable on endosomes in non-transfected cells, overexpression of mCh-MT1-MMP led to its endosomal hyper-recruitment (Fig. 1b). The MT1-MMP-dependent endosomal hyper-recruitment was apparent in both MDA-MB-231 and HeLa cells co-expressing TKS5-GFP and mCh-MT1-MMP (Fig. 1c, Supplementary Fig. 1a). The TKS5 homolog TKS4 showed a similar behaviour (Fig. 1d, Supplementary Fig. 1b). These results suggest that TKS4/5 are recruited to endosomes by MT1-MMP.

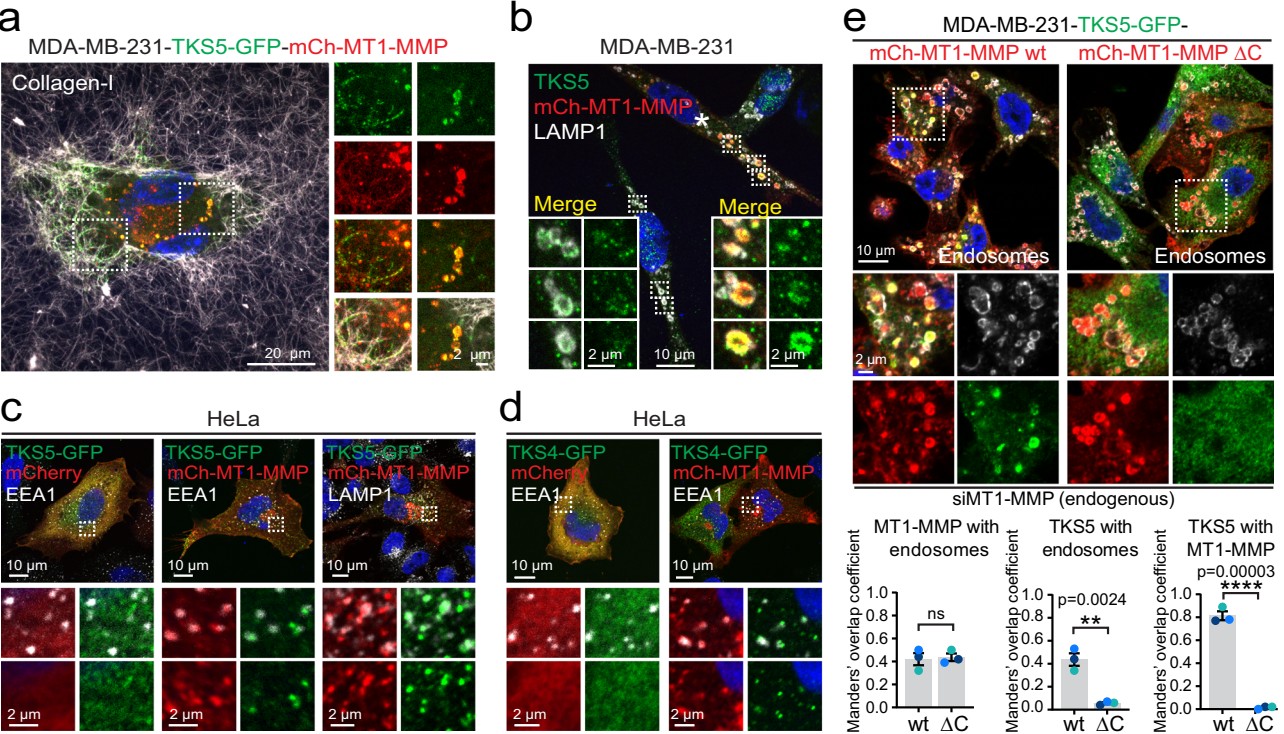

**Fig. 1 | TKS4/5 are recruited to endosomes by MT1-MMP. a** MDA-MB-231 cells stably expressing TKS5-GFP and mCh-MT1-MMP (MDA-MB-231-TKS5-GFP-mCh-MT1-MMP) were seeded on a thin layer of Alexa-647 labelled collagen-I, fixed and examined by confocal microscopy. Shown is a maximum intensity projection of a z-stack. TKS5-GFP and mCh-MT1-MMP co-localise both in linear invadopodia (left) and on endosome-like structures (right). Representative of 27 images from three experiments. **b** MDA-MB-231 cells were transiently transfected with mCh-MT1-MMP, fixed and immunostained for TKS5, mCh-MT1-MMP and LAMP1. Shown is a maximum intensity projection of a confocal z-stack. The mCh-MT1-MMP expressing cell (indicated by *) shows increased recruitment of endogenous TKS5 to LAMP1-positive endosomes. Representative of 17 images from two experiments. **c** Confocal images of HeLa cells transiently transfected with TKS5-GFP and mCh or

mCh-MT1-MMP showing co-occurrence of MT1-MMP and TKS5-GFP at early (EEA1) and late (LAMP1) endosomes. Representative of 24 (left), 10 (middle) and 24 (right) transfected cells from three experiments. **d** Confocal images of HeLa cells transiently transfected with TKS4-GFP and mCh or mCh-MT1-MMP showing co-occurrence of MT1-MMP and TKS4-GFP at early (EEA1) endosomes. Representative of 9 (left) and 11 (right) transfected cells from three experiments. **e** MDA-MB-231-TKS5-GFP-mCh-MT1-MMP wt/ΔC cells were transfected with siRNA targeting endogenous MT1-MMP, stained with antibodies against GFP, mCh and a combination of EEA1 and LAMP1 (Endosomes), and analysed by confocal microscopy. Graphs represent Manders' overlap coefficient showing mean +/− SEM from three independent experiments indicated by coloured dots. In total, 15 images with 5–7 cells were analysed per condition. Unpaired two-sided *t*-test.

Since TKS4/5 are cytosolic proteins, we asked whether their endosomal recruitment depends on the cytosolic tail of MT1-MMP, which comprises only 20 amino acid residues. MDA-MB-231 cells stably expressing a C-terminal deletion mutant of mCh-MT1-MMP, "MT1-MMPΔC", showed a slightly reduced intensity of TKS5-GFP fluorescence on CD63-positive endosomes compared to a cell line expressing the full-length mCh-MT1-MMP (Supplementary Fig. 1c, d), pointing to a role of the cytosolic tail in the recruitment of TKS4/5 to endosomes. We reasoned that the modest difference between mutant and wild-type (wt) mCh-MT1-MMP might be due to the presence of endogenous full-length MT1-MMP. Indeed, depletion of endogenous MT1-MMP completely abrogated TKS5-GFP localisation to mCh-MT1-MMPΔC positive endosomes, while mCh-MT1-MMP was sufficient for endosome recruitment of TKS5-GFP (Fig. 1e, Supplementary Fig. 1e). Similarly, mCh-MT1-MMPΔC failed to recruit TKS4-GFP or TKS5-GFP in HeLa cells (Supplementary Fig. 1f, g), which do not express endogenous MT1-MMP (Supplementary Fig. 1h). We conclude that the cytosolic tail of MT1-MMP is necessary for the recruitment of TKS4/5 to endosomes.

### Endosome recruitment of TKS4/5 depends on their functional PX domains

To investigate which region of TKS5 is required for this recruitment, we generated deletion mutants of TKS5 (Supplementary Fig. 2a) and cotransfected these together with mCh alone, mCh-MT1-MMP wt or ΔC in HeLa cells. HeLa cells neither express MT1-MMP, nor endogenous full length TKS5α, making them highly suitable for this type of analysis (Supplementary Fig. 1h). Scoring of the subcellular localisation by confocal microscopy revealed that the PX domain of TKS5 is necessary for the MT1-MMP-dependent endosomal localisation (Fig. 2a).

PX domains can interact with cellular membranes by binding to various lipids. Mutation of two key residues arginine 42 "R42" and arginine 93 "R93" (Fig. 2b) in the lipid-binding pocket has been shown to abrogate phosphoinositide binding for TKS5[9]. In agreement with published data[9,26–29], the wt PX domain of TKS5 showed a relatively promiscuous binding pattern to various phosphoinositides in a protein-lipid overlay assay, and the lipid binding was abrogated by double mutation of R42/R93 into alanines (Supplementary Fig. 2b). Interestingly, the single mutation of R93A affected lipid binding only marginally, whereas R42A completely abolished lipid binding of the PX domain.

To test the effect of the PX mutations for the MT1-MMP-dependent endosome recruitment, we generated single and double mutant full-length expression constructs of TKS4 and TKS5 (Supplementary Fig. 2c). The double mutation (μμ) or mutating only R94 in TKS4 (or R93 in TKS5) completely abrogated endosome localisation following mCh-MT1-MMP overexpression (Supplementary Fig. 2d). In contrast, mutation of the lipid-binding residue R43 in TKS4 (or R42 for TKS5) led to almost indistinguishable endosomal recruitment compared with the wt constructs in this readout (Supplementary Fig. 2d). We conclude that MT1-MMP-dependent endosome recruitment of TKS5 and TKS4 requires a functional PX domain and in particular R94 in TKS4 or R93 in TKS5, respectively.

### The PX domain of TKS4/5 interacts with the cytosolic tail of MT1-MMP

In addition to their well-known function as phosphoinositide binders, PX domains can also engage in protein-protein interactions[30,31]. To investigate whether the hyper-recruitment of TKS4/5 to endosomes following co-expression of MT1-MMP occurs through interaction between these proteins, we performed co-immunoprecipitation (Co-IP) experiments. TKS5-GFP co-immunoprecipitated mCh-MT1-MMP when cells were grown in the presence or absence of collagen-I (Supplementary Fig. 3a). In contrast, the C-terminal deletion of MT1-MMP did not interact with TKS5-GFP (Fig. 2c, Supplementary Fig. 3b), which is in line with the endosome recruitment data. Since R93 was

dispensable for lipid binding, but required for endosome localisation, we hypothesised that this residue might be engaged in protein binding. Indeed, mutation of R93, but not R42, abrogated the interaction with MT1-MMP in the context of the full-length TKS5 or when the PX domain alone was used for Co-IP experiments (Fig. 2d, Supplementary Fig. 3c). Taken together, this indicates that R93 is the residue responsible for MT1-MMP-dependent endosome recruitment of TKS5, whereas the lipid binding R42 residue appears to play a minor role.

In order to address whether the interaction between TKS5 and MT1-MMP is direct, we utilized the purified PX domain of TKS5 (Supplementary Fig. 2b) in an MBP pull-down experiment with the purified C-terminal 20 amino acids of MT1-MMP as a GST-fusion peptide. Our attempts to show an in vitro interaction between both purified proteins failed, possibly because the purified MT1-MMP-C-terminus does not fold correctly or because the interaction with the PX domain of TKS5 may require membrane association or the contribution from another protein. Importantly, however, purified MBP-TKS5-PX-wt, but not MBP-TKS5-PX-μμ, showed an interaction with endogenous MT1-MMP and mCh-MT1-MMP from cell lysates (Fig. 2e, Supplementary Fig. 3d).

### MT1-MMP-dependent endosome recruitment of TKS4/5 requires PtdIns3P

Since the R42 residue in TKS5 was crucial for lipid binding but less important for MT1-MMP interaction, we asked whether lipid binding through the PX domain contributes to the MT1-MMP-mediated endosomal recruitment of TKS5. Quantification of fluorescence intensities revealed a small reduction of TKS5-R42A on endosomes compared to wt TKS5 (Supplementary Fig. 4a). Depletion of endosomal PtdIns3P by use of the PI3K class III/VPS34 inhibitor SAR405[32], significantly reduced the amount of TKS5 and TKS4 in MT1-MMP dots (Fig. 2f, Supplementary Fig. 4b). Acute perfusion with SAR405 during live cell imaging, resulted in a reduction of TKS5-GFP fluorescence from mCh-MT1-MMP-positive endosomes within few minutes (Supplementary Fig. 4c and Supplementary Movie 1), which is in line with the observed dynamics of PtdIns3P reporter proteins, such as mCh-2xFYVE[33]. When comparing the effect of SAR405 on endosomal versus invadopodia-localised TKS5-GFP pools, only the endosomal TKS5-GFP responded to SAR405 (Supplementary Fig. 4d and Supplementary Movie 2). This was expected, since TKS5 localisation to invadopodia depends on PtdIns(3,4)P$_2$ at the plasma membrane[9,34], which is not affected by SAR405 treatment. The lipid binding ability of the PX domain to PtdIns3P thus contributes to the endosomal localisation of TKS4/5.

### MT1-MMP and TKS5 are internalized into intraluminal vesicles of late endosomes

The transmembrane protein MT1-MMP can be internalised into multivesicular endosomes (MVEs)[35], and we asked whether the cytosolic TKS5 would follow due to the interaction with the C-terminal tail of MT1-MMP. Since MDA-MB-231 cells have relatively large endosomes, confocal imaging allows to distinguish the limiting membrane from the endosomal lumen. Indeed, TKS5-GFP and TKS4-mNG appeared to localise inside endosomes together with mCh-MT1-MMP (Fig. 3a, Supplementary Fig. 5a). We verified that the endosomes, which are double positive for both mCh-MT1-MMP and TKS5-GFP, were MVEs by correlative light and electron microscopy (Supplementary Fig. 5b). Moreover, immunoelectron microscopy allowed visualization of mCh-MT1-MMP and TKS5-GFP inside the lumen of MVEs, indicating that TKS5 indeed follows MT1-MMP into intraluminal vesicles (ILVs) of MVEs (Fig. 3b, Supplementary Fig. 5c).

### MT1-MMP facilitates secretion of TKS4 and TKS5 in extracellular vesicles

Since MT1-MMP has been shown to be secreted in exosomes[20], we wondered whether MVE-internalized TKS4/5 might be co-secreted in

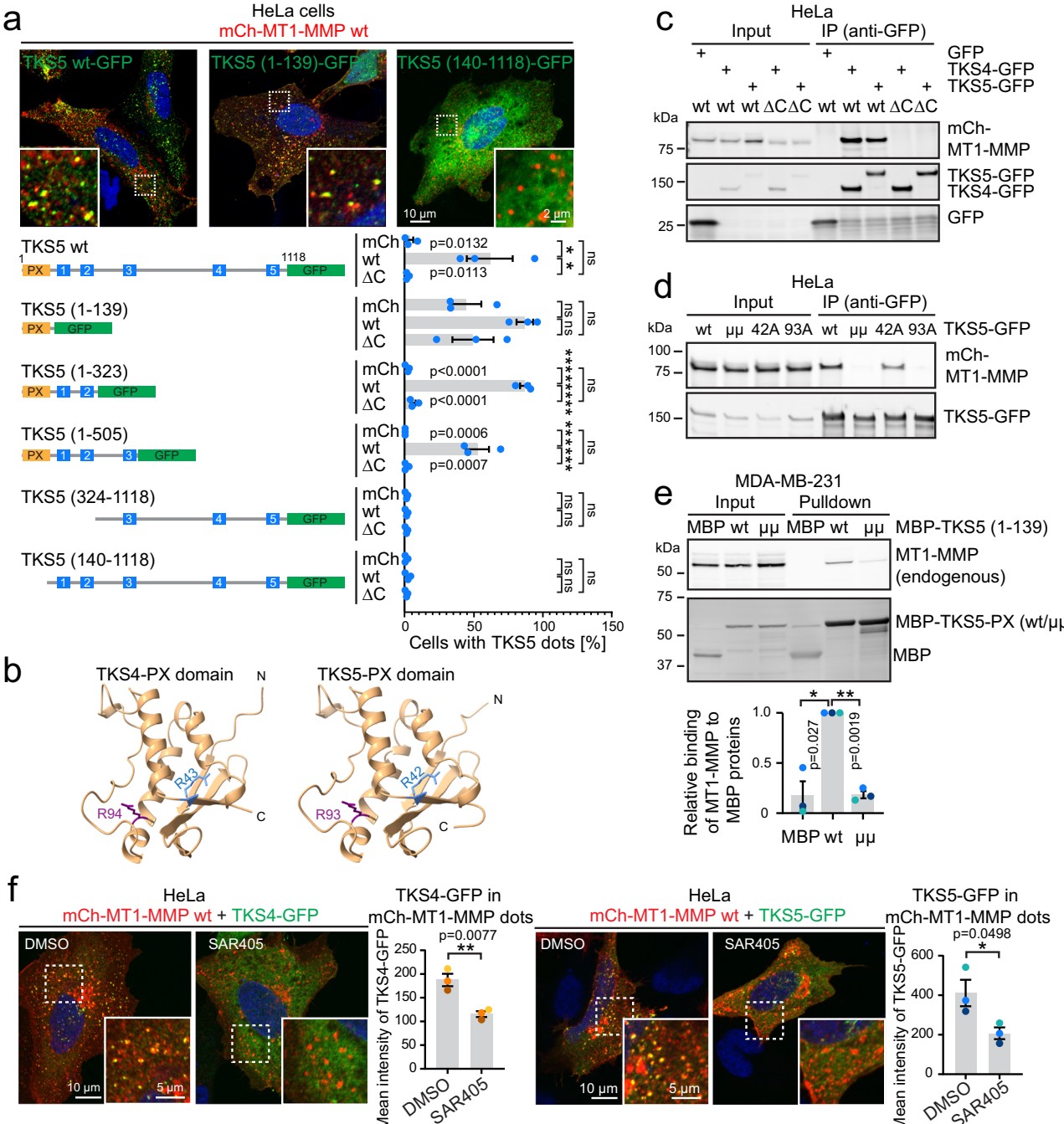

**Fig. 2 | TKS4/5 are recruited to endosomes by coincidence binding of their PX domain to MT1-MMP and PtdIns3P. a** HeLa cells were transfected with mCh or mCh-MT1-MMP wt or ΔC and cotransfected with the indicated TKS5 constructs. The localisation of TKS5 was scored by fluorescence microscopy and example images with and without endosomal recruitment are displayed. Graph shows the percentage of cells with a dot-like localisation of the respective TKS5 construct as mean +/− SEM from n = 3 independent experiments. At least 100 cells were counted per condition and experiment. One-way ANOVA of each individual construct set of 3 conditions with Tukey's multiple comparisons test. **b** Predicted 3D protein structure of the PX domains from human TKS5 and TKS4 using AlphaFold2 via ColabFold. Arginine residues R42/R43 and R93/94 are highlighted. **c** Co-immunoprecipitation experiment using HeLa cells transfected with TKS5-GFP and mCh-MT1-MMP wt or ΔC. mCh-MT1-MMP wt, but not ΔC can be co-immunoprecipitated with TKS4- or TKS5-GFP. Representative of three independent experiments. **d** Co-immunoprecipitation experiment using HeLa cells transfected with mCh-MT1-MMP wt and TKS5-GFP wt, R42A, R93A or the double mutant R42A/

R93A ("μμ"). mCh-MT1-MMP wt can be co-immunoprecipitated with TKS5-GFP wt and the R42A mutant, but not when R93 is mutated. Representative of three independent experiments. **e** Mannose-binding-protein (MBP) pulldown experiment. Purified MBP-TKS5 PX wt, but not MBP-TKS5 PX μμ binds to MT1-MMP from MDA-MB-231 cell lysate. Purified MBP alone serves as a negative control. Graph shows the relative mean intensity of the MT1-MMP band quantified from n = 3 independent experiments +/− SEM. One-sample two-sided t-test. **f** HeLa cells were co-transfected with mCh-MT1-MMP wt and TKS4- or TKS5-GFP and treated with DMSO or 6 μM SAR405 for 15 min, stained with antibodies against GFP and mCh and analysed by confocal microscopy. Graphs represent the mean fluorescence intensity of TKS4-GFP or TKS5-GFP in mCh-MT1-MMP dots, measured automatically by NIS-Elements. Error bars denote mean +/− SEM from n = 3 independent experiments indicated by coloured dots. TKS4-GFP: In total 60 (DMSO) and 53 (SAR405) cells were analysed. TKS5-GFP: In total, 60 cells were analysed per condition, unpaired two-sided t-test.

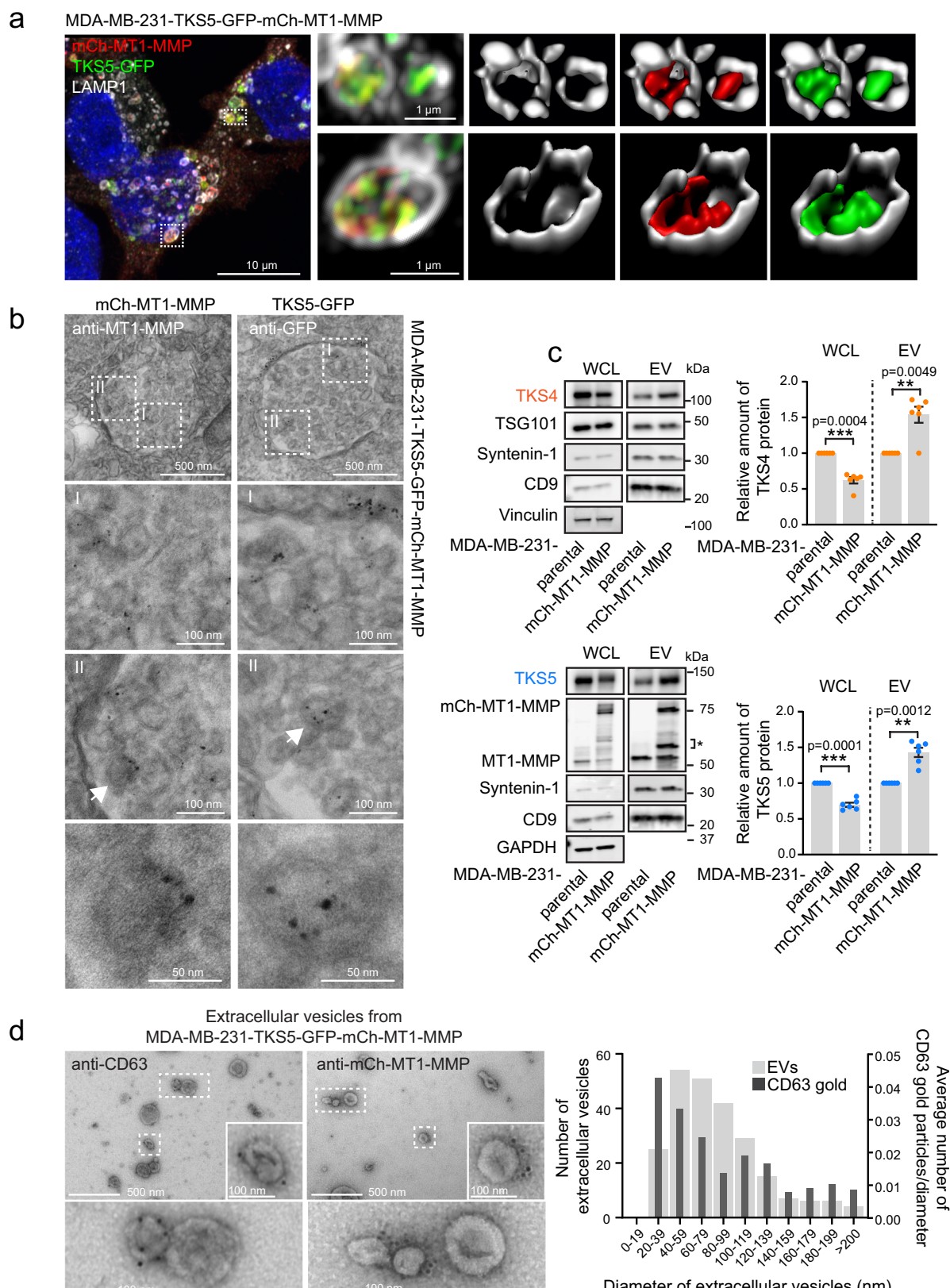

MT1-MMP-positive exosomes. In support of this, we observed that protein levels of TKS4 and TKS5 were reduced in cells overexpressing mCh-MT1-MMP wt but not ΔC (Fig. 3c, Supplementary Fig. 6a). Moreover, TKS4/5 mRNA levels were slightly increased in MT1-MMP overexpressing cells presumably to compensate for the loss of TKS4/5 (Supplementary Fig. 6b). Importantly, overexpression of mCh-MT1-

MMP increased the levels of endogenous TKS4 and TKS5 in extracellular vesicles (EVs) that were isolated by ultracentrifugation (Fig. 3c). To verify that the increased amount of TKS4/5 in the EV preparation was not due to an increase in apoptotic bodies from cells overexpressing mCh-MT1-MMP, we investigated cell death in both cell lines. We did not detect differences in caspase 3/7 activity or in the

**Fig. 3 | TKS4/5 and MT1-MMP are internalized into intraluminal vesicles of late endosomes and co-secreted in extracellular vesicles. a** MDA-MB-231-TKS5-GFP-mCh-MT1-MMP cells were grown on cover slips, fixed and immunostained with antibodies against TKS5, mCh and LAMP1. Samples were analysed by super-resolution microscopy (Airyscan). Maximum intensity projection of a z-stack and Imaris surface 3D renderings are shown, visualizing that TKS5-GFP and mCh-MT1-MMP appear to localise inside endosomes. Representative of 10 images of 19 cells. **b** Immunogold labelling with silver enhancement of MDA-MB-231-TKS5-GFP-mCh-MT1-MMP cells with antibodies against MT1-MMP or GFP. Electron microscopy images show labelling of mCh-MT1-MMP (left column) and TKS5-GFP (right column) inside the lumen of MVEs. The arrows point to immunogold-labelling-positive ILVs. Representative of 21 (anti-MT1-MMP) and 28 (anti-GFP) MVEs. **c** MDA-MB-231 parental or mCh-MT1-MMP stably over-expressing cells were used to isolate extracellular vesicles (EVs) using ultracentrifugation (100 000xg) as described in the methods section. Western blots (WB) and graphs show the presence of the indicated proteins in whole cell lysate (WCL) normalized to vinculin or GAPDH, and EV fraction normalized to TSG101 (TKS4) and to CD9 (TKS5). * MT1-MMP variants likely comprising processed forms of MT1-MMP and mCh-MT1-MMP (immature, posttranslationally modified or cleaved). Error bars denote mean +/− SEM from $n = 6$ independent experiments. One sample two-sided $t$-test. **d** EVs were isolated from MDA-MB-231-TKS5-GFP-mCh-MT1-MMP cells as described in c, and prepared for electron microscopy as described in the methods. Immunogold labelling using antibodies against mCherry or CD63 show labelling on the surface of the EVs. Graph represents the diameter distribution and the corresponding density of CD63 labelling of in total 239 EVs collected from three independent experiments.

amount of cleaved PARP-1 as readouts for apoptosis between the two cell lines (Supplementary Fig. 6c, d). Moreover, MT1-MMP over-expression did not seem to affect the amount of EVs as judged by the EV markers TSG101, Syntenin-1 and CD9 (Fig. 3c).

Characterization of the extracellular vesicle fraction by immunoelectron microscopy showed vesicles of various sizes, with 72% being below 100 nm, consistent with the size of intraluminal vesicles of MVEs. These vesicles labelled positive for CD63 and mCh-MT1-MMP (Fig. 3d). TKS5-GFP was not detected on the EVs, consistent with a localisation in the lumen of the EVs, making it inaccessible to antibody labelling.

Taken together, we conclude that MT1-MMP facilitates secretion of TKS4/5 in EVs most likely of endosomal origin.

### TKS4/5 promote non-autocatalytic cleavage of MT1-MMP by bridging ADAM sheddases with MT1-MMP

Since we found that TKS4/5 were recruited by MT1-MMP to endosomes and EVs, we asked whether MT1-MMP was affected by the interaction with the TKS4/5 proteins. Intriguingly, we noticed that overexpression of TKS4-GFP or TKS5-GFP in HeLa cells co-transfected with mCh-MT1-MMP induced a significant increase in the amount of a 34 kDa mCh-positive band in the whole cell lysate compared to cells without TKS4/5 overexpression (Fig. 4a, b, Supplementary Fig. 7a). The 34 kDa mCh-positive band likely corresponded to the membrane bound fragment generated by non-autocatalytic cleavage of mCh-MT1-MMP, which releases a 50−52 kDa catalytically active ectodomain[22,23,36–38] (Fig. 4c). The 34 kDa fragment was absent in cells expressing the C-terminal 20 amino acid deletion of MT1-MMP-mCherry (Fig. 4a,b, Supplementary Fig. 7a). Moreover, overexpression of the TKS4 PX double mutant, incapable of interacting with MT1-MMP, was not able to promote the 34 kDa band (Fig. 4d, Supplementary Fig. 7b). Thus, our results indicate that overexpression of TKS4 or TKS5 promotes non-autocatalytic cleavage of MT1-MMP and that this requires its cytosolic tail and a functional TKS4/5-PX domain. Inversely, knockdown (KD) of TKS4 in HeLa cells reduced the intensity of the 34 kDa band and this could be rescued by overexpression of an siRNA-resistant TKS4-GFP construct (Fig. 4e, Supplementary Fig. 7c). Thus, TKS4/5 can regulate the efficiency of non-autocatalytic cleavage of MT1-MMP in HeLa cells by interacting with its cytosolic tail via their PX domains.

We next asked whether this type of cleavage could also be observed in MDA-MB-231 cells stably expressing mCh-MT1-MMP. Indeed, we identified the 34 kDa band in whole cell lysate and EVs (Supplementary Fig. 7d), and we detected the corresponding soluble 50-52 kDa ectodomain from a spin column purification of enriched conditioned medium (CM), using an antibody against the catalytic domain of MT1-MMP (Fig. 4f). Importantly, co-depletion of TKS4 and TKS5 virtually abolished the cleavage of MT1-MMP as visualized by a strong reduction of the 34 kDa and the 50-52 kDa bands in the whole cell lysate and the CM respectively (Fig. 4f, Supplementary Fig. 7e). The TKS4/5 dependent cleavage was also apparent in MDA-MB-231 cells stably expressing untagged MT1-MMP, as well as in endogenous MT1-MMP (Fig. 4g, Supplementary Fig. 7f–h). Taken together, we conclude that TKS4/5 facilitate non-autocatalytic cleavage of MT1-MMP.

Non-autocatalytic cleavage of MT1-MMP is known to be mediated by other metalloproteases, specifically from the ADAM family[22]. Indeed, when HeLa or MDA-MB-231 cells were treated with a dilution series of various metalloprotease inhibitors, the presence of the 34 kDa mCh-MT1-MMP fragment was reduced. Whereas overnight treatment with GM6001, which mainly inhibits MT1-MMP activity, only gave a minor decrease in the 34 kDa band, Marimastat gave a slightly higher effect, and especially Batimastat was very potent (Fig. 5a, Supplementary Fig. 8a). This inhibitor profile is consistent with the function of ADAM proteases[39], and in line with existing literature regarding non-autocatalytic processing of MT1-MMP[22,38].

Interestingly, TKS4 and TKS5 have been shown to interact with ADAM12, 15 and 19[9,10]. Indeed, co-expression of ADAM12-myc or ADAM15-myc-SNAP with mCh-MT1-MMP and TKS4/5-GFP revealed that these proteins co-localise on endosomes (Fig. 5b, Supplementary Fig. 8b). Moreover, using Co-IP ADAM15-myc-SNAP was preferentially found in a complex with mCh-MT1-MMP in the presence of wt, but not PX-mutant TKS4 or TKS5 (Fig. 5c, Supplementary Fig. 8c), indicating that TKS4/5 can bridge ADAM sheddases with MT1-MMP. Taken together, our results suggest that TKS4/5 can facilitate ADAM-mediated MT1-MMP processing, and that this depends on their ability to interact with the cytosolic tail of MT1-MMP.

### Cleavage of MT1-MMP occurs in acidic endosomes

Metalloprotease activity can be dependent on pH[40–42], begging the question whether MT1-MMP cleavage can be affected by the acidic endosomal pH. Indeed, deacidification of endosomes using $NH_4Cl$ reduced MT1-MMP shedding (Fig. 5d). Inversely, acidifying the cell culture medium slightly increased cleavage, albeit to a much lower degree (Supplementary Fig. 8d). These data indicate that ectodomain shedding occurs to a major extend in the acidic lumen of endosomes, in line with the prominent localisation of TKS4/5 in MT1-MMP positive endosomes.

### Conditioned medium mediates intercellular transfer of cancer cell invasiveness

We next asked whether the CM from MDA-MB-231 cells, which contains EVs and the shed ectodomain of MT1-MMP, might display degradative activity towards ECM proteins. Therefore, we coated coverslips with fluorescently labelled gelatin and added CM from MDA-MB-231 cells ("donor cells") stably expressing mCh-MT1-MMP and TKS5-GFP. No degradation of gelatin was observed under these conditions (Fig. 6a). Since EVs can communicate with cells[43], we addressed whether the presence of cells on the fluorescent gelatin would trigger the degradative potential of the CM. As recipient cells, we used HeLa cells or MCF7 cells, which do not express MT1-MMP and have poor invasive properties (Supplementary Fig. 9a)[20,44,45]. While the recipient cells alone were unable to degrade gelatin, addition of CM from two different MT1-MMP positive donor cell lines equipped the recipient cells

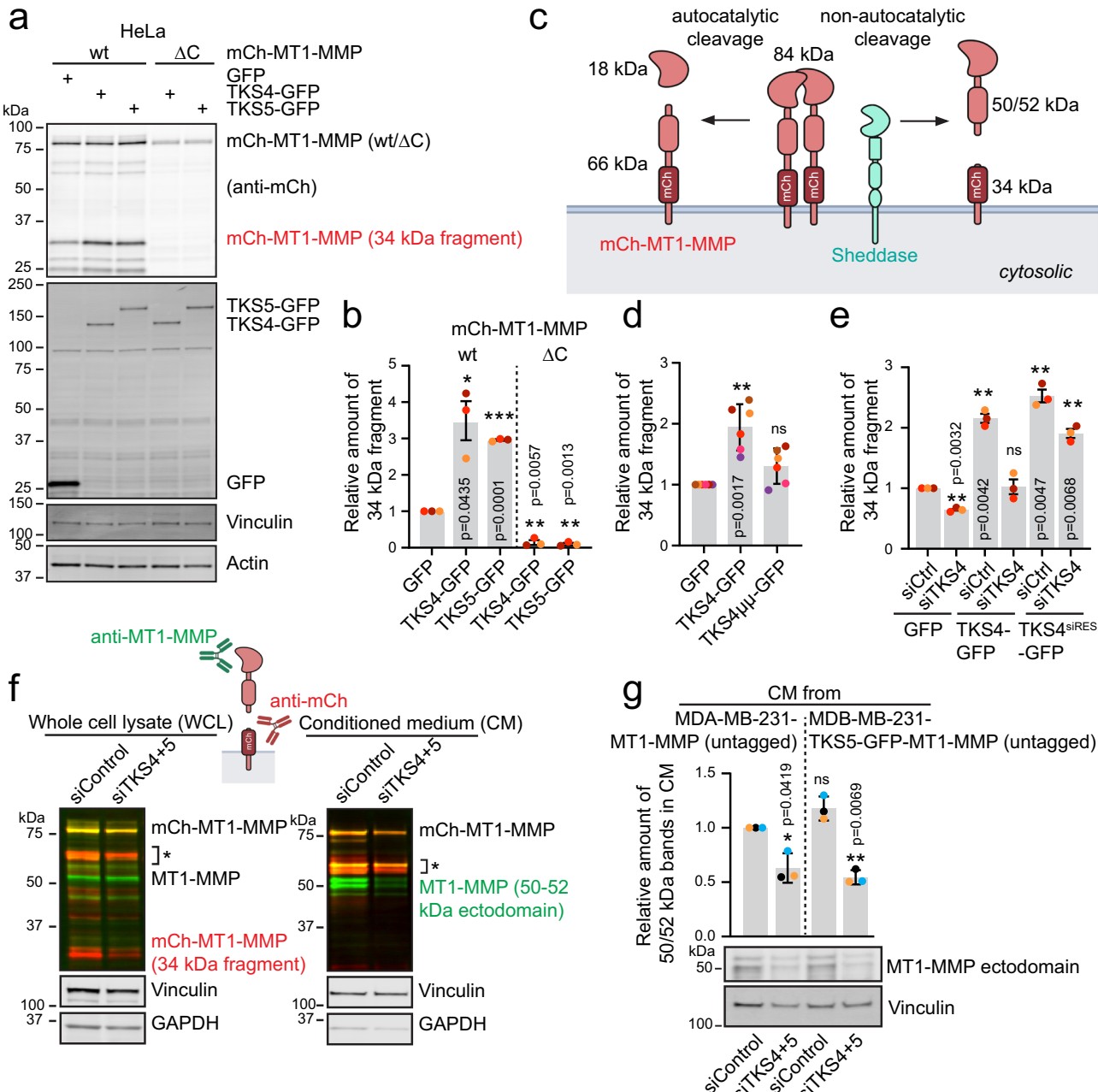

**Fig. 4 | TKS4/5 promote shedding of MT1-MMP. a** HeLa cells were co-transfected with TKS4-GFP or TKS5-GFP together with mCh-MT1-MMP wt or ΔC, lysed and subjected to WB. **b** Quantification of WB from (**a**) representing the relative intensities of the 34 kDa cleavage product of MT1-MMP from *n* = 3 independent experiments, mean +/− SEM. One sample two-sided *t*-test. **c** Illustration showing autocatalytic and non-autocatalytic cleavage of mCh-MT1-MMP and the resulting fragments. Created with BioRender.com. **d** HeLa cells were co-transfected with mCh-MT1-MMP wt and GFP, TKS4-GFP wt or the double mutant of TKS4 (μμ). Total cell lysates were analysed by WB, one representative WB is shown in Supplementary Fig. 7b. The graph shows the relative intensities of the 34 kDa cleavage product of MT1-MMP. Data is from *n* = 6 independent experiments, mean +/− SD. One sample two-sided *t*-test. **e** Rescue experiment: HeLa cells were depleted for TKS4 and then transfected with GFP, TKS4-GFP or an siRNA resistant version of TKS4-GFP (TKS4-GFP^siRES) together with mCh-MT1-MMP. Depletion of TKS4 reduces the amount of cleaved MT1-MMP, while overexpression of TKS4 increases it (one representative WB is shown in Supplementary Fig. 7c). An siRNA resistant construct of TKS4

rescues the siRNA-mediated effect on cleavage. The graph shows the relative intensities of the 34 kDa cleavage product of MT1-MMP. Data is from *n* = 3 independent experiments, mean +/− SEM. One sample two-sided *t*-test. **f** MDA-MB-231-TKS5-GFP-mCh-MT1-MMP wt cells were transfected with siRNA targeting TKS4 and TKS5 for 4 days. The whole cell lysate (WCL) and the spin-column concentrated conditioned medium (CM) were analysed by WB with anti-mCh (red) and anti-MT1-MMP-catalytic domain (green) antibodies. Representative of 4 WB. * MT1-MMP variants likely comprising processed forms of MT1-MMP and mCh-MT1-MMP (immature, posttranslationally modified or cleaved). Illustration created with BioRender.com. **g** MDA-MB-231-MT1-MMP (untagged) or MDA-MB-231-TKS5-GFP-MT1-MMP (untagged) cells were transfected with siRNA targeting TKS4 and TKS5 for 4 days. The spin-column concentrated CM was analysed by WB with an anti-MT1-MMP-catalytic domain antibody. Graph shows the quantification of the 50–52 kDa band from CM of the indicated cell lines. Data from *n* = 3 independent experiments, mean +/−SD. One sample two-sided *t*-test.

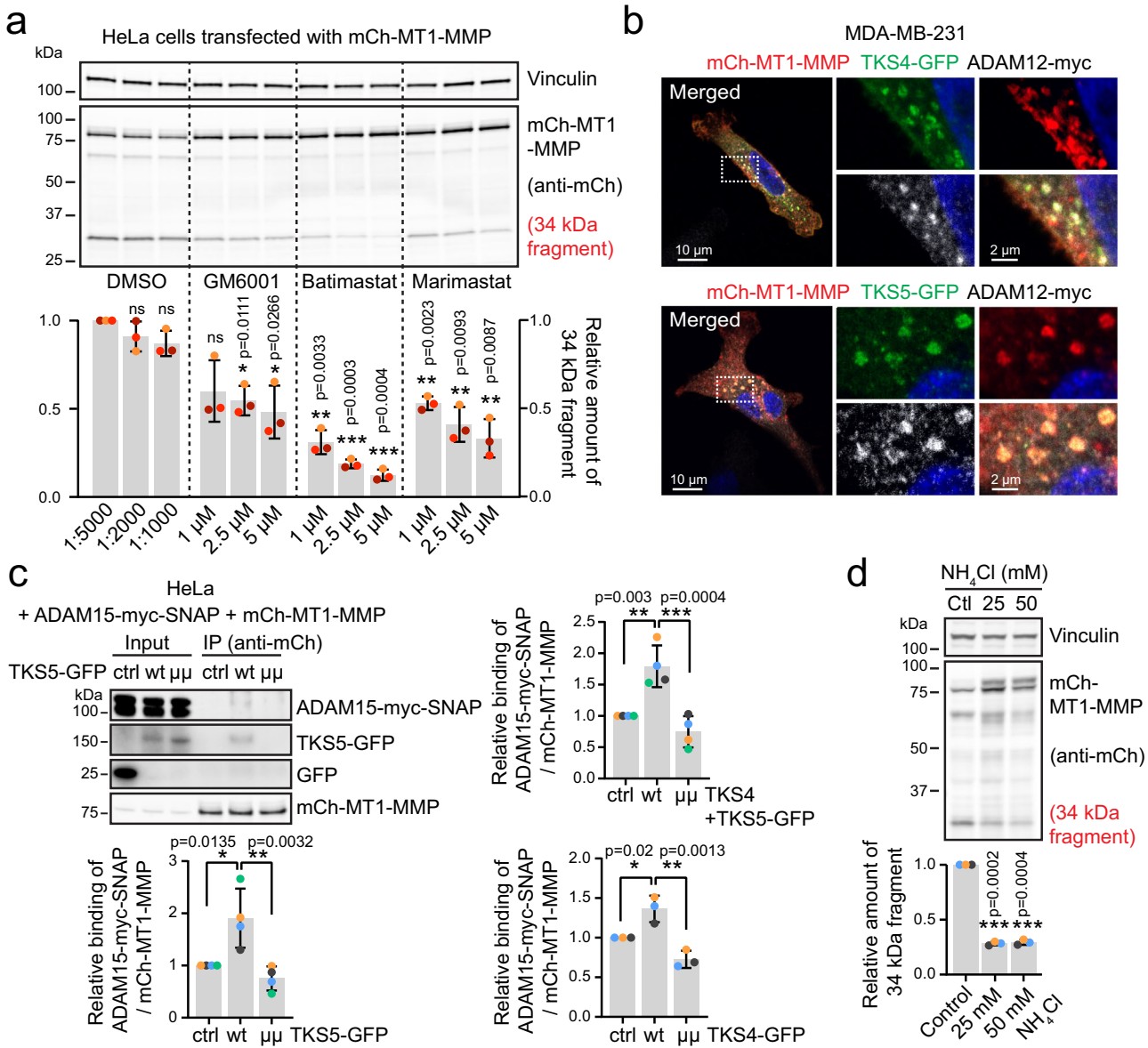

**Fig. 5 | TKS4/5 promote ADAM-mediated shedding of MT1-MMP in acidic endosomes. a** HeLa cells were transfected with mCh-MT1-MMP. 3 h post transfection, cells were treated with metalloproteinase inhibitors as indicated. Protein lysates were made 24 h after the transfection and analysed by WB. Graph shows the quantification of the 34 kDa cleavage product of MT1-MMP. Cells treated with the lowest concentration of DMSO were set to 1. Data is from n = 3 independent experiments, mean +/− SD. One sample two-sided t-test. **b** MDA-MB-231 cells were transiently transfected with mCh-MT1-MMP, TKS4- or TKS5-GFP and ADAM12-myc. 24 h after transfection, cells were fixed and stained with anti-mCh, anti-GFP and anti-Myc antibodies and analysed by confocal microscopy. mCh-MT1-MMP, TKS4/5-GFP and ADAM12-myc colocalise on endosomes. Displayed is a maximum intensity projection of 2 slices à 0.9 µm section thickness. Representative of 13 (TKS4) and 6 (TKS5) confocal image acquisitions. See also Supplementary Fig. 8b for

ADAM15. **c** Co-immunoprecipitation experiment using HeLa cells transfected with mCh-MT1-MMP, ADAM15-myc-SNAP and TKS5-GFP wt or µµ or GFP (ctrl). ADAM15-myc-SNAP preferentially co-immunoprecipitates with mCh-MT1-MMP in the presence of wt, but not mutated TKS5-GFP ("µµ") or GFP alone. Graphs show the relative mean intensity +/−SD of the ADAM15-myc-SNAP signal quantified from cells transfected with TKS5-GFP (n = 4), TKS4-GFP (n = 3) or a combination of both (n = 4). Representative WBs for TKS4-GFP and TKS4 + 5-GFP Co-IP experiments are shown in Supplementary Fig. 8c. One-way ANOVA with Tukey's multiple comparisons test. **d** MDA-MB-231-TKS5-GFP-mCh-MT1-MMP cells were treated with 25 or 50 mM NH₄Cl for 20 h. Total cell lysates were analysed by WB. Graph shows the quantification of the 34 kDa cleavage product of MT1-MMP. Data is from three independent experiments, mean +/− SD, one sample two-sided t-test.

with the ability to degrade gelatin, as indicated by black cell-associated areas in the labelled gelatin (Fig. 6a–c, Supplementary Fig. 9a). Moreover, addition of the CM made HeLa cells invasive when embedded in collagen-I, and HeLa spheroids embedded in collagen-I expanded in the presence of CM (Fig. 6d, Supplementary Fig. 9b, c + Supplementary Movie 3).

In order to establish a more physiological environment, we next addressed whether MT1-MMP positive cells had the ability to directly

transfer their invasive properties in a co-culture with non-invasive cells. Indeed, when HeLa cells were co-cultured with MDA-MB-231 cells stably expressing mCh-MT1-MMP and TKS5-GFP in serum free medium, they gained the ability to invade into a matrix of collagen-I towards an increasing gradient of chemo attractants (Fig. 6e). Taken together, these data indicate that the MT1-MMP positive cells can transfer their degradative and invasive properties to MT1-MMP negative cells through the extracellular environment.

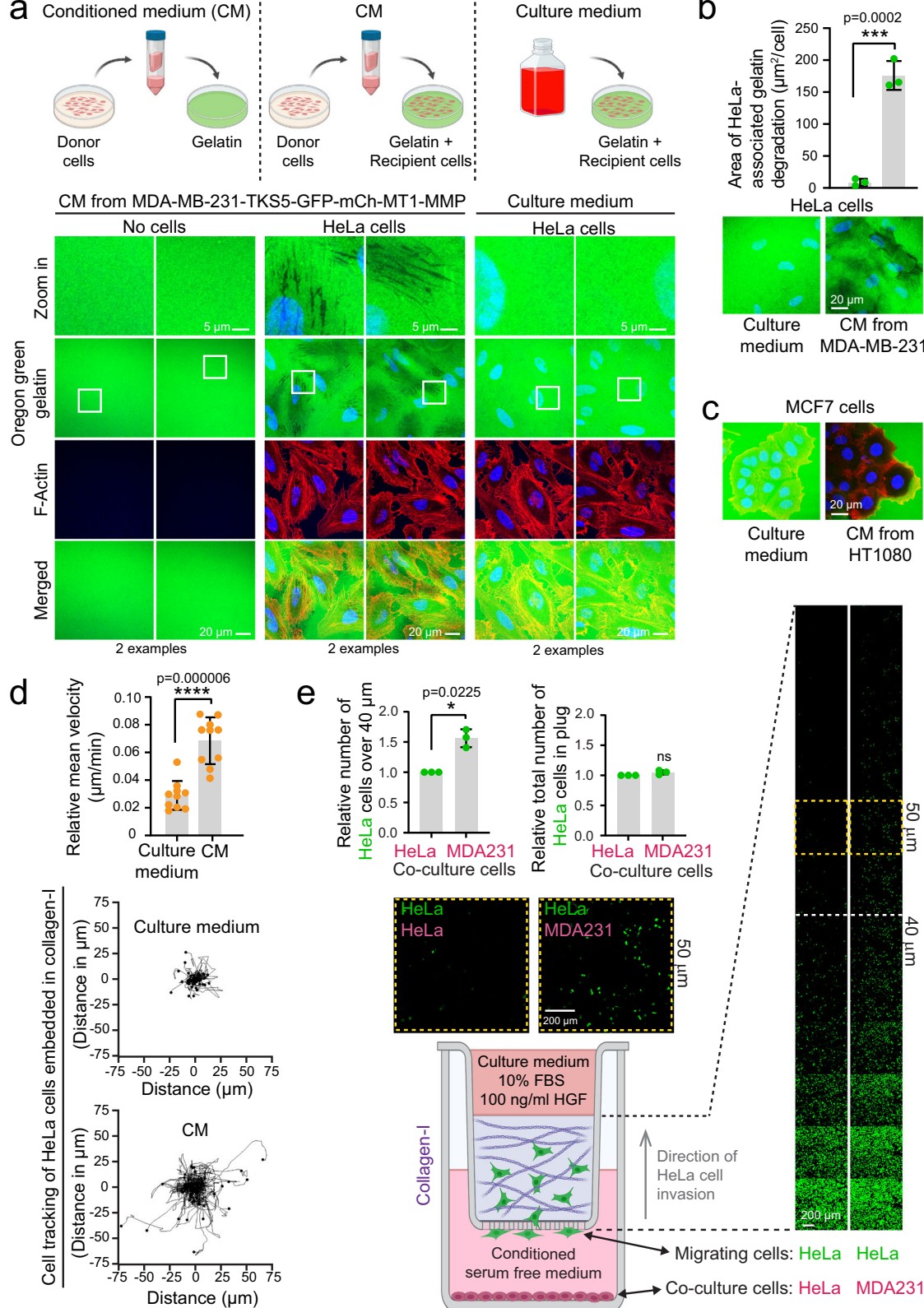

## MT1-MMP from donor cells interacts with recipient cells

The CM-induced gelatin degradation was greatly reduced when the recipient cells were fixed with formaldehyde prior to CM-stimulation, indicating that the components of the CM actively communicate with the recipient cells (Fig. 7a). The gelatin degradation surrounding the recipient HeLa cells partly followed actin stress fibres (Fig. 7b),

suggesting that components of the CM could interact with the HeLa cells. By fluorescence confocal microscopy, we could detect mCh-MT1-MMP from the CM of MDA-MB-231 cells on the surface of HeLa cells (Fig. 7c, Supplementary Fig. 10a), in association with Zyxin-labelled focal adhesions (Fig. 7d, Supplementary Fig. 10b) and even in HeLa endosomes (Fig. 7e, Supplementary Fig. 10c). Accordingly, mCh-MT1-

**Fig. 6 | Conditioned medium from invasive cells makes non-invasive cells degradative and invasive. a** HeLa cells were seeded on Oregon Green™ 488 labelled gelatin and treated with conditioned medium (CM) from MDA-MB-231-TKS5-GFP-mCh-MT1-MMP cells or culture medium for 24 h. Cells were labelled with rhodamine-Phalloidin (actin) and Hoechst 33342 (nuclei). Representative of 10 images per condition. Illustration created with BioRender.com. **b** HeLa cells were seeded on Oregon Green™ 488 labelled gelatin and treated with conditioned medium from MDA-MB-231 cells or regular cell culture medium for 24 h. Graph represents the area of gelatin degradation per cell, measured automatically by NIS-Elements and shown as mean +/− SD from $n = 3$ independent experiments. In total >300 cells were analysed per condition. Unpaired two-sided $t$-test. **c** MCF7 cells were seeded on Oregon Green™ 488 labelled gelatin and treated with CM from HT1080 cells or regular cell culture medium for 24 h. Representative of 25 images per condition from two independent experiments. **d** HeLa cells embedded in type I collagen were incubated with culture medium or CM from MDA-MB-231-TKS5-GFP-mCh-MT1-MMP cells and analysed by live cell microscopy. Graph shows the relative

mean velocity per cell ($\mu$m min$^{-1}$) +/−SD. Dots represent the mean value from $n = 10$ wells (each comprising 35–80 cells) collected from three independent experiments. Cells tracked in total, culture medium: 367, CM: 497. Unpaired two sided $t$-test. Rose plots represent cell tracks from one well per condition. Culture medium 55 cells; CM 58 cells. **e** Inverted invasion assay of HeLa cells into a collagen-I matrix. Co-culturing with invasive (MDA-MB-231-TKS5-GFP-mCh-MT1-MMP) cells stimulates invasive migration of HeLa cells (labelled with Calcein AM, green) when compared to co-culturing them with non-invasive cells (HeLa). Graphs represent the relative number of cells in the collagen plugs (right) and the amount of cells which invaded more than 40 $\mu$m into the collagen-I matrix (left). Error bars denote mean +/− SD from $n = 3$ independent experiments, each comprising three collagen plugs with 5–6 confocal z-stacks per condition. One-sample two-sided $t$-test. Example images from one representative image stack per condition are displayed. Images marked with a yellow frame are magnifications of the 50 $\mu$m z slice. Illustration created with BioRender.com.

MMP was detected in cell lysate from HeLa cells treated with CM from MDA-MB-231 cells (Fig. 7f). Intriguingly also the 50-52 kDa shed ectodomain of MT1-MMP was found in cell lysate from the CM-treated HeLa cells (Fig. 7f). This indicates that mCh-MT1-MMP-positive EVs from MDA-MB-231 cells can interact with HeLa cells and that the shed catalytic ectodomain of MT1-MMP is able to associate with HeLa cells, either directly or via the EVs.

The association of EVs and MT1-MMP with the recipient cells could potentially trigger transcriptional responses. The mRNA levels of TKS4/5 or MT1-MMP were, however, not increased in HeLa cells treated with CM (Supplementary Fig. 10d).

We next addressed whether uptake of material by endocytosis was important for the degradative capabilities of the recipient cells. Inhibition of clathrin mediated endocytosis in HeLa cells did not affect their ability to degrade gelatin (Fig. 7g, Supplementary Fig. 10e, f). However, when global endocytic uptake was inhibited by hypertonic treatment[46], the HeLa-associated gelatin degradation was significantly reduced (Fig. 7h, Supplementary Fig. 10g). This indicates that the CM-induced transfer of invasiveness is more efficient when the recipient cells are capable of performing endocytic uptake. This could be due to changes in endocytic trafficking and transcytosis of internalised MT1-MMP in the HeLa recipient cells[47].

### Intercellular transfer of invasiveness requires functional endocytic trafficking in the donor cell

In order to characterize the invasive potency of the CM in more detail, we next separated the CM in two fractions, enriched for either EVs or soluble proteins. The EV markers CD9 and ALIX were only detected in the EV fraction, whereas the soluble metalloprotease MMP2 was enriched in the soluble fraction (Fig. 8a). The 50–52 kDa MT1-MMP ectodomain was enriched in the soluble fraction, but was also found in the EV fraction. This suggests that the 50–52 kDa ectodomain can either be fully soluble or associate with the EVs. Importantly, both fractions were equally potent in cell-associated gelatin degradation (Fig. 8a). This shows that the soluble fraction containing the 50–52 kDa MT1-MMP ectodomain can instruct an ECM-degradative phenotype in the absence of EVs.

We next asked whether the formation of multivesicular endosomes was required for MT1-MMP shedding and transfer of invasiveness. To inhibit ILV formation, cells were co-depleted for HRS and CD63 (Supplementary Fig. 11a), which mediate ESCRT (endosomal sorting complex required for transport)-dependent and CD63-dependent ILV formation, respectively[48,49]. This led to enlarged endosomes, with reduced luminal mCh-MT1-MMP and TKS5-GFP, in line with impaired ILV formation (Fig. 8b). Accordingly, the amount of the EV marker Syntenin-1 was reduced in the conditioned medium (Fig. 8c), whereas the 50–52 kDa shed MT1-MMP ectodomain was still present (Fig. 8d). This indicates that cleavage of MT1-MMP and release

of the ectodomain can occur in the absence of ILV formation, presumably by fusion of ILV-devoid endosomes with the plasma membrane. Importantly, the conditioned medium from HRS and CD63 co-depleted cells transferred degradative properties to HeLa cells on gelatin, although to a lesser extent than control treated cells (Fig. 8e). This is consistent with the finding from the previous fractionation experiment (Fig. 8a), and shows that the soluble MT1-MMP ectodomain has degradative properties in the absence of endosome-derived EVs.

We next asked whether late endosome translocation and function was important for MT1-MMP trafficking and transfer of invasiveness. The ER protein Protrudin mediates Kinesin-1-dependent translocation of MVEs towards the plasma membrane[50], whereas the late endosomal small GTPase RAB7 is crucial for late endosome maturation and function[51]. siRNA-mediated depletion of Protrudin led to perinuclear clustering of endosomes as expected, whereas RAB7-depletion induced the formation of enlarged aberrant endosomes, which accumulated mCh-MT1-MMP and TKS5-GFP (Fig. 8b, Supplementary Fig. 11b, c). In both conditions, the amount of Syntenin-1 and the 50–52 kDa ectodomain in the conditioned medium, were reduced (Fig. 8c, d). This shows that proper function and transport of late endosomes are required to release EVs and the MT1-MMP ectodomain. In line with this, conditioned medium from RAB7 depleted cells was virtually unable to induce HeLa-associated gelatin degradation (Fig. 8e).

Taken together, our results show that both EVs and the shed MT1-MMP ectodomain can mediate transfer of invasiveness, that they are more potent together, and that the presence of both in the conditioned medium depends on functional late endosomes.

### Intercellular transfer of invasiveness depends on TKS4/5 and MT1-MMP in the donor cell

The CM-induced invasion in collagen-I was strongly reduced upon GM6001 treatment, indicating that it was metalloprotease dependent (Supplementary Fig. 12a). To test whether the ECM-degradation was mediated by MT1-MMP, we depleted MDA-MB-231 or HT1080 cells for MT1-MMP by siRNA and added CM from these cells to HeLa- or MCF7 cells seeded on gelatin. Indeed, recipient cell-associated gelatin degradation was severely impaired in MT1-MMP KD samples (Fig. 9a–c, Supplementary Fig. 12b, c). Since the CM did not significantly increase the ability of the recipient cells to migrate in general (Supplementary Fig. 12d), our results suggest that the gained invasive properties of the recipient cells depend on MT1-MMP from the donor cells.

Next, we addressed whether the presence of TKS4 and TKS5 in the donor cells would affect the ECM-degradative potency of the CM. For this purpose, we depleted MDA-MB-231-TKS5-GFP-mCh-MT1-MMP cells for TKS4 and TKS5, alone or in combination and collected the CM

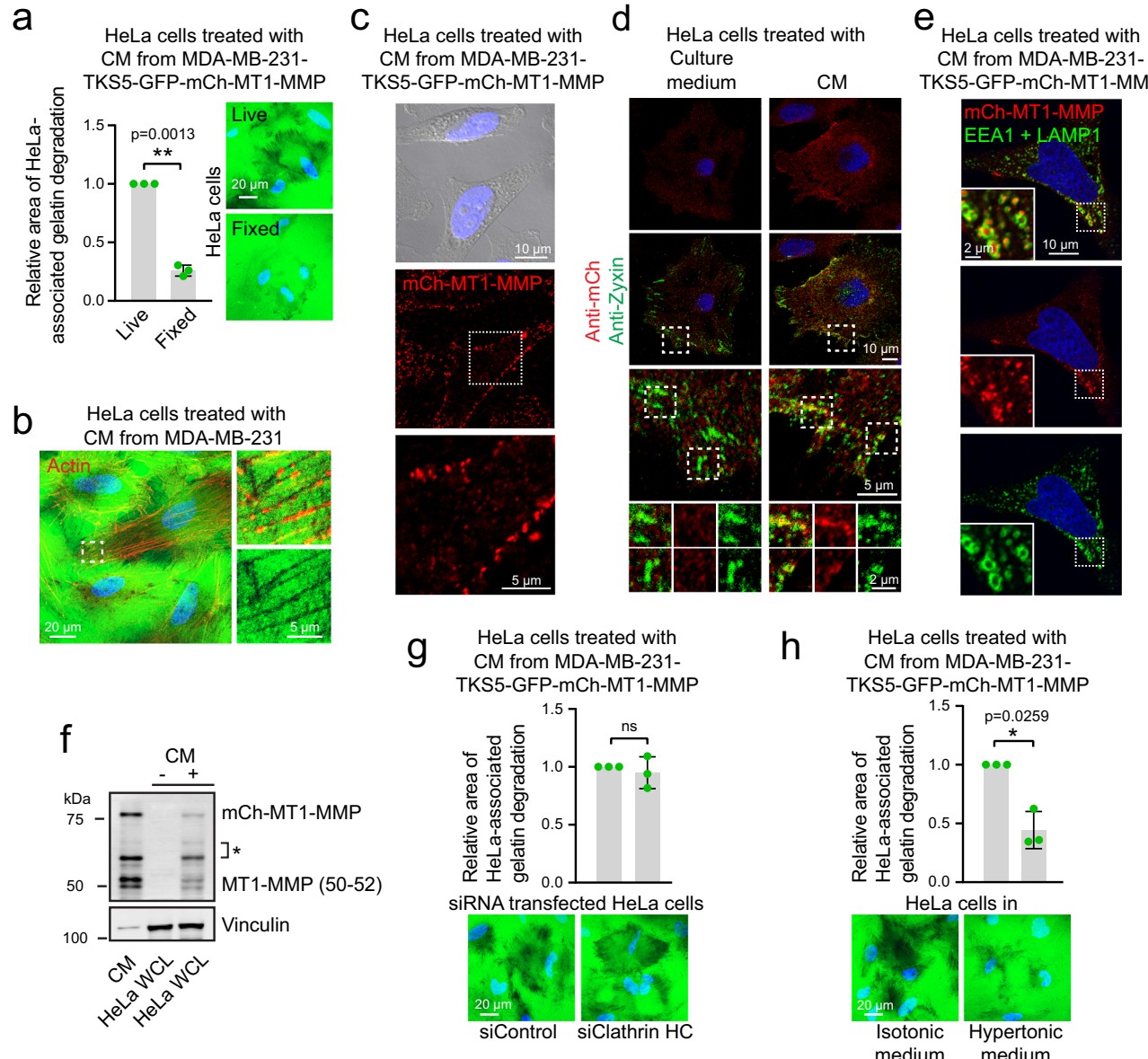

**Fig. 7 | MT1-MMP from the conditioned medium interacts with recipient cells important for gelatin degradation. a** Oregon Green™ 488 labelled gelatin was fixed in 3% formaldehyde before seeding of HeLa cells. After 4 h, the cells were fixed or not before addition of CM from MDA-MB-231-TKS5-GFP-mCh-MT1-MMP cells for 24 h. Graph represents the relative amount of the area of gelatin degradation per cell, mean +/− SD from *n* = 3 independent experiments. Live, 326 cells, Fixed, 333 cells, one-sample two-sided *t*-test. **b** HeLa cells on Oregon Green™ 488 gelatin treated with CM for 24 h. Areas of degraded gelatin (dark) partly overlap with actin (Alexa568Phalloidin). Representative of 12 images from two independent experiments. **c** Detection of mCh-MT1-MMP on the surface of HeLa cells treated with CM for 2 h. Representative of 18 images from two independent experiments. **d** HeLa cells seeded on fibronectin, treated with CM from MDA-MB-231-TKS5-GFP-mCh-MT1-MMP or culture medium for 2 h and stained with anti-mCh and anti-Zyxin (focal adhesion marker). Representative of 22 images from three independent experiments. **e** Detection of mCh-MT1-MMP in endosomes of HeLa cells treated

with CM for 2 h. Representative of 50 images from three independent experiments. **f** WB of CM from MDA-MB-231-TKS5-GFP-mCh-MT1-MMP cells (left lane) and WCL from HeLa cells treated or not with CM for 4 h. mCh-MT1-MMP full length and the 50–52 kDa ectodomain can be detected in lysate from the CM-treated HeLa cells. * MT1-MMP variants likely comprising processed forms of MT1-MMP and mCh-MT1-MMP. Representative of three WB. **g** HeLa cells depleted or not for clathrin heavy chain (HC) were seeded on Oregon Green™ 488 gelatin and treated with CM for 24 h. Graph represents the relative area of gelatin degradation per cell, mean +/− SD from *n* = 3 independent experiments. In total >290 cells were analysed per condition. One sample two-sided *t*-test. **h** HeLa cells seeded on Oregon Green™ 488 gelatin were incubated with isotonic or hypertonic (addition of 100 mM NaCl) CM for 24 h. Graph represents the relative area of gelatin degradation per cell, mean +/− SD from *n* = 3 independent experiments. In total >300 cells were analysed per condition. One sample two-sided *t*-test.

(Supplementary Fig. 12e). TKS5 depletion has been shown to reduce invadopodia-associated exosome secretion[52]. The content of total EVs in the CM, measured by the EV markers TSG101 and CD81, was, however, not affected by TKS4/5 depletion in our experimental setup (Supplementary Fig. 12f). While the individual KDs showed marginal effects on CM-induced gelatin degradation by HeLa cells, the depletion of TKS4 together with TKS5 strongly decreased degradation and

invasion (Fig. 9d, e), indicating that TKS4/5 can compensate for each other. In support of this, endogenous TKS4 was upregulated in TKS5 depleted cells (Supplementary Fig. 12e). These data show that the concerted action of TKS4 and TKS5 has a pronounced effect on the ECM-degradative and invasive potency of the CM. The TKS4/5-dependent transfer of degradative properties was confirmed avoiding overexpression of MT1-MMP or TKS proteins in MDA-MB-231 and

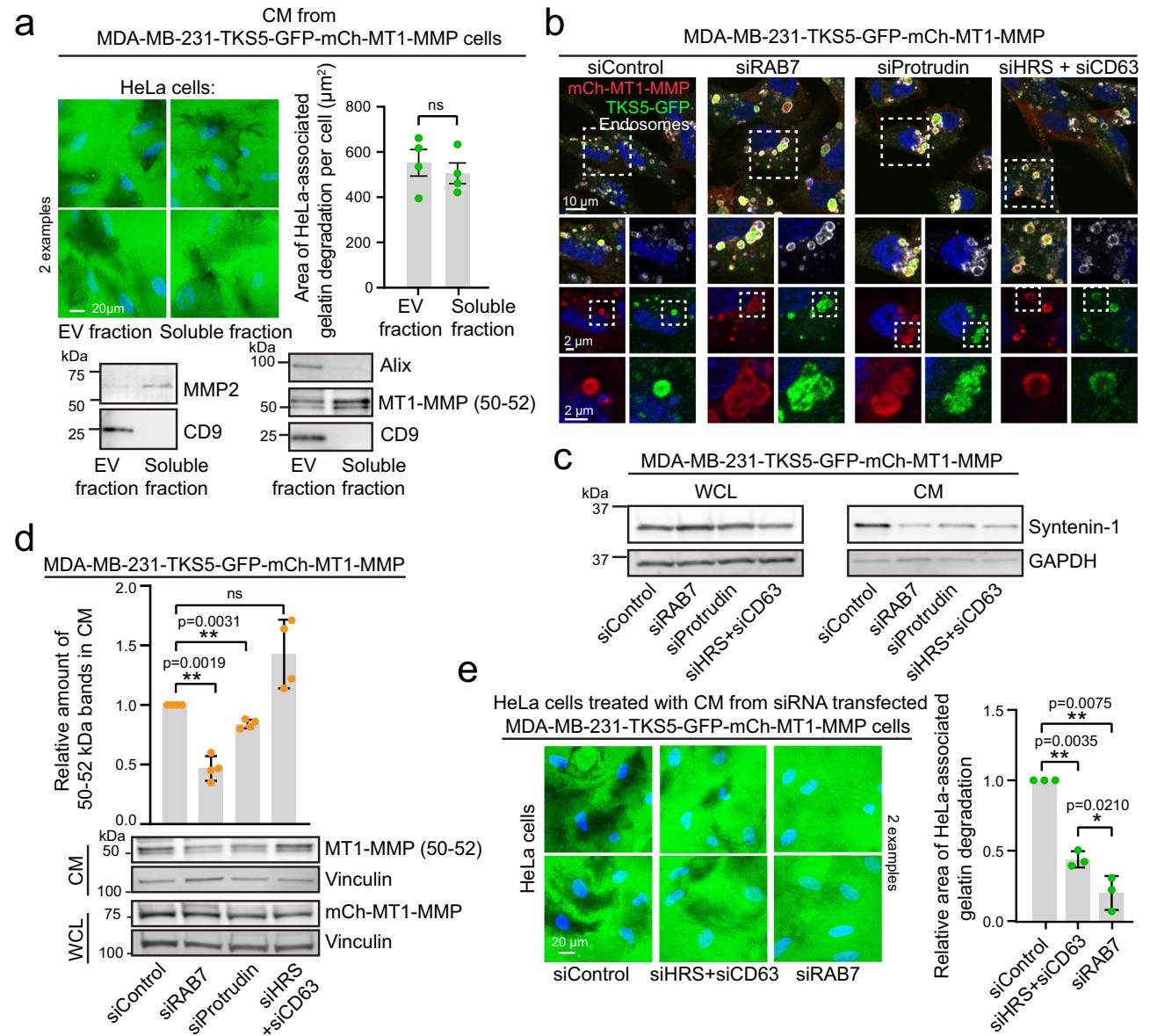

**Fig. 8 | EVs and the MT1-MMP ectodomain transfer invasive properties depending on endocytic trafficking in the donor cell. a** HeLa cells were seeded on Oregon Green™ 488 labelled gelatin and treated with CM for 24 h. The CM was concentrated using a 100 K spin column to enrich for extracellular vesicles (EV fraction). The flow through was concentrated using a 10 K spin column to enrich for soluble proteins (soluble fraction). Graph represents the area of gelatin degradation per cell, mean +/− SEM from $n = 4$ independent experiments. In total >400 cells were analysed per condition. Unpaired two-sided *t*-test. WB of the conditioned medium showing enrichment of the EV markers CD9 and Alix, and the soluble marker MMP2 in the indicated fractions. Note that the 50–52 kDa MT1-MMP ectodomain is found in both fractions, but enriched in the soluble fraction. **b** MDA-MB-231-mCh-MT1-MMP-TKS5-GFP cells were depleted for the indicated proteins by siRNA and analysed by confocal microscopy using antibodies against mCh, GFP and a mixture of EEA1 and LAMP1 to visualize endosomes. Representative of 15 images per condition from three independent experiments. **c** WB showing how KD of endosomal proteins reduces the amount of Syntenin-1 positive EVs in CM. Representative of four WB. **d** WB and quantification of the level of the 50–52 kDa MT1-MMP ectodomain in CM from siRNA transfected MDA-MB-231-mCh-MT1-MMP-TKS5-GFP cells. Graph shows mean +/− SD from $n = 4$ independent experiments, one-sample two-sided *t*-test. **e** HeLa cells were seeded on Oregon Green™ 488 labelled gelatin and treated for 24 h with CM from siRNA transfected MDA-MB-231-TKS5-GFP-mCh-MT1-MMP cells. Graph represents the relative amount of the area of gelatin degradation per cell, mean +/− SD from $n = 3$ independent experiments. In total >300 cells were analysed per condition. One sample two sided *t*-test (relative to control) and one way-ANOVA with Tukey's multiple comparisons test (HRS + CD63/RAB7).

HT1080 cells (Fig. 9f, g, Supplementary Fig. 12g, h). Indeed, the extent of HeLa-induced ECM degradation correlated closely with the amount of the 50-52 kDa MT1-MMP ectodomain in the CM from TKS4/5 depleted MDA-MB-231 cells (Fig. 9h). Moreover, the enzymatic activity of the 50-52 kDa products was significantly reduced in TKS4/5-depleted cells as measured by Zymography (Fig. 9i). This suggests that TKS4/5 promote transfer of cancer cell invasiveness through the generation of the catalytically active 50−52 kDa ectodomain.

We conclude that TKS4/5 promote ectodomain shedding of MT1-MMP, thereby enabling ECM degradation associated with cells at

distant sites from MT1-MMP expressing cells. MT1-MMP and TKS4/5 are frequently overexpressed in cancer, associated with tumour progression, metastases and poor survival[53–57]. The TKS4/5- and MT1-MMP-dependent transfer of invasiveness shown here increases our understanding of these proteins in cancer development.

## Discussion

Overexpression of MT1-MMP and TKS4/5 promotes cancer progression, ascribed to their function in invadopodia biogenesis[1,8]. Here, we identify an additional, invadopodia-independent, mechanism for how

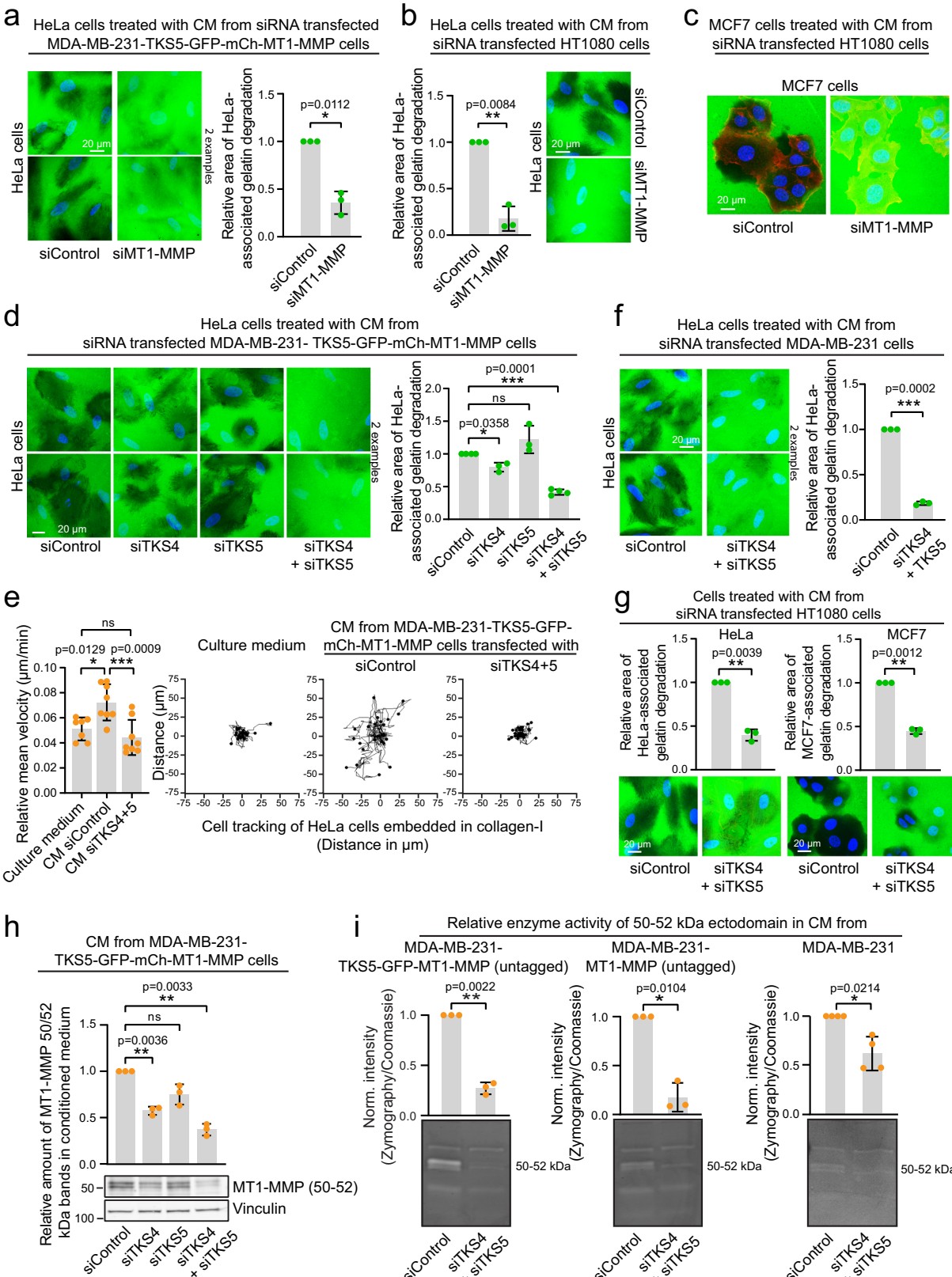

these proteins promote cancer progression: In acidic endosomes, TKS4/5 facilitate ADAM-mediated cleavage of MT1-MMP by bridging the two proteases. Subsequent endosome fusion with the plasma membrane releases the catalytically active MT1-MMP-ectodomain into the extracellular space, where it docks on the surface of other cells. By

this mechanism, previously non-invasive cells gain neo-invasive properties (Fig. 10).

TKS4 and TKS5 are indispensable for invadopodia formation. These adaptor proteins are anchored to PtdIns(3,4)P₂ rich areas of the plasma membrane via their PX domain, where they interact with actin

**Fig. 9 | TKS4/5 and MT1-MMP in the donor cell define the potency of the conditioned medium. a–d** Recipient cells seeded on Oregon Green™ 488 gelatin treated with CM from siRNA treated donor cells for 24 h. Graphs in (**a**) and (**b**) represent the relative area of gelatin degradation per cell, mean +/− SD from $n = 3$ independent experiments. In total >300 cells were analysed per condition. One sample two-sided $t$-test. Images in (**c**) are representative of 6 images per condition. Graph in (**d**) represents the relative area of gelatin degradation per cell, mean +/− SD. siControl: $n = 4$ (455 cells), siTKS5: $n = 3$ (355 cells), siTKS4: $n = 3$ (336 cells), siTKS4 + TKS5: $n = 4$ (464 cells). One sample two-sided $t$-test. **e** HeLa cells embedded in type I collagen were incubated with culture medium or CM from cells that had been depleted or not for TKS4/5 and analysed by live cell microscopy. Graph shows the relative mean velocity per cell ($\mu$m min$^{-1}$) +/−SD. Dots represent the mean value of individual wells (each comprising 75–170 cells), from three independent experiments. Culture medium: $n = 7$ wells (679 cells), CM siControl: $n = 8$ wells (969 cells), CM siTKS4/5: $n = 8$ wells (913 cells). One-way ANOVA, Tukey's multiple comparisons test. Rose plots represent cell tracks from one movie per condition. Culture medium: 24 cells, CM siControl: 26 cells, CM siTKS4/5: 27 cells. **f**, **g** Recipient cells seeded on Oregon Green™ 488 gelatin treated with CM from siRNA treated donor cells for 24 h. Graph in (**f**) represents the relative area of gelatin degradation per cell, mean +/− SD from $n = 3$ independent experiments. In total >300 cells were analysed per condition. One sample two-sided $t$-test. Graphs in (**g**) represent the relative area of gelatin degradation per cell, mean +/− SD from $n = 3$ independent experiments. In total >250 cells (HeLa) and >450 cells (MCF7) were analysed per condition. One sample two-sided $t$-test. **h** Graph and WB showing the relative amount of the mCh-MT1-MMP 50-52 ectodomain in CM. Mean +/−SD, $n = 3$ independent experiments, One sample two-sided $t$-test. **i** Zymography gels and quantification of enzyme activity of the 50–52 kDa MT1-MMP ectodomain in the indicated cell lines. Graphs represent mean normalized intensity of 50–52 kDa bands, +/− SD from $n = 3$ (left and middle) or $n = 4$ (right) independent experiments. One sample two-sided $t$-test.

regulators and cortactin through their SH3 domains, thereby promoting actin polymerization and branching important for invadopodia growth and maturation[8,58]. Here we find that TKS4/5 are recruited to endosomal membranes via their PX domain, by a coincidence detection of endosomal PtdIns3P and the cytosolic tail of MT1-MMP. This provides a rationale for the previously unexplained dual specificity of the TKS4/5 PX domain for both PtdIns3P and PtdIns(3,4)P$_2$[9,27]. TKS5 has previously been detected on endosomes[59,60], and TKS4 has been observed in cytoplasmic dots resembling endosomes[27]. However, the functional role of endosomal TKS4/5 has not been known. Now, we show that TKS4/5 promote ADAM-mediated shedding of MT1-MMP in acidic endosomes. TKS4/5 proteins have been found to interact with several members of the ADAM family of sheddases via their C-terminal SH3 domains[9,10,61,62]. We find that TKS4/5 can bridge ADAM proteases to MT1-MMP by use of their N-terminal PX domains. In support of this, PX mutants of TKS4/5 unable to interact with MT1-MMP failed to promote shedding.

The pH-dependent cleavage of MT1-MMP confines this processing to acidic endosomes and provides a level of spatiotemporal control of the catalytically active MT1-MMP-ectodomain. Upon fusion of multivesicular endosomes with the plasma membrane the ectodomain was released, together with EVs that contained TKS4/5 and MT1-MMP. Interestingly, ADAM sheddases and MT1-MMP have previously been reported to be secreted in exosomes[63] and shedding can occur from exosome membranes[18,64]. Our results suggest that the presence of TKS4/5 in EVs could facilitate such MT1-MMP processing, thereby providing a sustained release mechanism of MT1-MMP from EVs. This is supported by the notion that the tumour microenvironment is often acidic[65]. We cannot rule out that the TKS4/5-dependent shedding can additionally occur from the plasma membrane. In support of this, we occasionally observed TKS4/5 recruitment to the MT1-MMP-positive plasma membrane, presumably recruited by a coincidence detection of MT1-MMP and PtdIns(3,4)P$_2$[9,34]. Indeed, acidification of the cell culture medium slightly increased MT1-MMP shedding. However, deacidifying endosomes had a much more pronounced effect on cleavage, indicating that substantial MT1-MMP processing occurs within endosomes.

Importantly, the shed ectodomain and MT1-MMP-containing EVs associated with recipient HeLa cells after treatment with the CM from MT1-MMP expressing donor cells. This converted the previously MT1-MMP-negative HeLa cells into ECM-degradative and invasive cells. Interestingly, CM alone was not sufficient to induce gelatin degradation, which strictly required the presence of cells. We assume that association of EVs and/or the ectodomain to the cell surface allows a confined and prolonged contact of the metalloprotease with its substrate (Fig. 10).

In addition to residing at the cell surface of HeLa cells, MT1-MMP-containing EVs were internalised into HeLa endosomes, contributing to efficient transfer of degradative properties. These EVs might be degraded in lysosomes, or the endosomes might recycle and fuse with the plasma membrane releasing its MT1-MMP containing EVs at a different site. In an analogous way, EVs from a donor cell have been shown to trigger TLR9 signalling in endosomes of recipient cells, resulting in increased endosome transport to the cell periphery and increased ECM degradation[47]. The MT1-MMP-positive endosomes in CM-treated HeLa cells might be subjected to a similar regulation, allowing targeted secretion of internalised MT1-MMP and thereby potentiating the ECM-degradative effect of the CM.

Directed transport of MT1-MMP-positive endosomes to invadopodia and fusion with the invadopodial plasma membrane is required for invadopodia maturation and ECM degradation[6,16]. Our finding that MT1-MMP recruits TKS4/5 to endosomes suggests that TKS4/5 might be co-transported with MT1-MMP to growing invadopodia. This would provide an alternative and targeted route for localising TKS proteins to forming invadopodia, in concert with MT1-MMP. There, TKS4/5 could facilitate the shedding of MT1-MMP, and in addition contribute to the formation of invadopodia. Moreover, exosome secretion is enhanced by invadopodia, thereby enhancing the invasive phenotype[52]. Thus, targeted transport of endosomes to invadopodia is crucial for efficient ECM degradation, and the endosomal presence of TKS4/5 and MT1-MMP may contribute to this function in several ways.

We have shown here that the recruitment of TKS adaptor proteins to MT1-MMP-positive endosomes promotes MT1-MMP's ability to degrade ECM at distant sites. This is partly explained by our finding that TKS4/5 facilitate the non-autocatalytic cleavage of its catalytically active ectodomain. We cannot exclude that the TKS proteins can influence MT1-MMP activity also in other ways. They could regulate its activity towards soluble proteases like MMP2 and MMP9 and/or affect MT1-MMP trafficking and surface expression[13]. Moreover, TKS proteins have been implicated in the production of reactive oxygen species (ROS)[66], which is known to activate MMPs[67]. Indeed, the enzymatic activity of MT1-MMP as measured by zymography was higher in the presence of TKS4/5 as compared to CM collected from TKS4/5 depleted donor cells, which could imply a direct activation of enzyme activity through TKS proteins. However, since the enzyme activity corresponded roughly to the amount of secreted ectodomain, MT1-MMP cleavage likely represents the main mechanism for TKS-dependent transfer of invasiveness.

Overexpression of MT1-MMP and TKS proteins promotes cancer progression in various cancer types[1,15,68]. These proteins are known to function in parallel, having independent but equally important roles in promoting invadopodia formation and function. Our finding that TKS4/5 interact with MT1-MMP in endosomes and EVs provides a new understanding of the tight cooperation of these proteins in ECM degradation, which will be important for our understanding of cancer progression and metastasis.

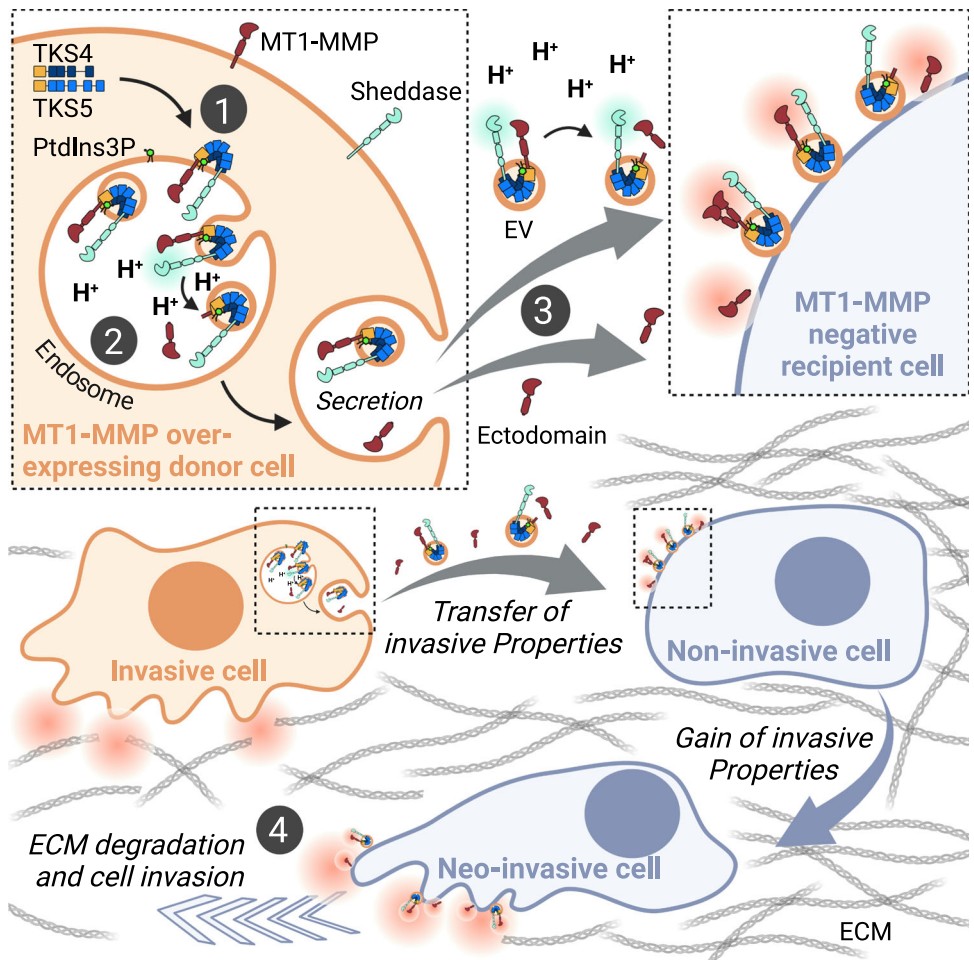

**Fig. 10 | Model of intercellular transfer of invasiveness through endosomal protease shedding.** Summarizing model: TKS4/5 are recruited to endosomes by binding to the endosomal lipid PtdIns3P and the cytosolic tail of MT1-MMP, and subsequently sorted into intraluminal vesicles of multivesicular endosomes in concert with MT1-MMP (1). TKS4/5 bridges MT1-MMP with members of the ADAM family of sheddases and the low endosomal pH facilitates ADAM-mediated cleavage of a catalytically active ectodomain of MT1-MMP in the endosome lumen (2). MT1- MMP-containing exosomes and its shed ectodomain are secreted by donor cells and dock onto MT1-MMP negative, non-invasive recipient cells (3). This induces degradation of the extracellular matrix surrounding the recipient cell. Consequently, the recipient cell becomes invasive (4). The presence of TKS4/5 in EVs could facilitate sustained MT1-MMP shedding in the acidic tumour microenvironment[65]. Created with BioRender.com.

## Methods

### Antibodies and reagents

Antibodies used in this study were obtained from the following resources: Goat anti-mCherry was from Acris (AB0040-200, WB 1:1000, Immuno-EM 1:50). Human anti-EEA1 serum[69] (IF 1:160000) was a gift from Ban-Hock Toh, Melbourne, Australia. Rabbit anti-LAMP1 (L1418, IF 1:400), mouse anti-β-actin (A5316, WB 1:5000), mouse anti-vinculin (V9131, WB 1:3000), rabbit anti-TKS5 (HPA037923, IF 1:100, WB 1:1000-1:2000), rabbit anti-TKS4 (HPA036471, WB 1:250), mouse anti-cortactin (05–180, IF 1:200), mouse anti-GFP (11814 460001, IF 1:500, WB 1:1000), mouse anti-MT1-MMP (MAB3328, IF 1:200, WB 1:500, Immuno-EM 1:100) and rabbit anti-Zyxin (HPA004835, IF 1:100) were from Merck/Sigma. The cMyc monoclonal antibody (9E10, IF undiluted) developed by J.M. Bishop, University of California, San Francisco, the mouse-anti-LAMP1 antibody (H4A3, deposited by August, J.T. / Hildreth, J.E.K., IF 1:400), and the mouse-anti-CD63 antibody (H5C6, RRID:AB_528158, deposited by August, J.T. / Hildreth, J.E.K., IF 1:200, WB 1:500, Immuno-EM 1:50) were obtained from the Developmental Studies Hybridoma Bank, created by the NICHD of the NIH and maintained at The University of Iowa, Department of Biology, Iowa City, IA 52242. Mouse anti-MBP antibody (NEB E8032S, PIP strips™ 1:2000) was from New England Biolabs. Mouse anti-CD81

antibody (302-820, WB 1:500) was from Ancell Corporation. Rabbit anti-CD9 antibody (ab263019, WB 1:1000), rabbit anti-GFP (ab6556, Immuno-EM 1:500), rabbit anti-Syntenin-1 (ab133267, WB 1:1000), rabbit anti-clathrin heavy chain (ab21679, WB 1:1000) and mouse anti-GAPDH antibody (ab9484, WB 1:3000) were from Abcam. Mouse anti-TSG101 (612697, WB 1:500) was from BD Transduction Lab. Rabbit anti-PARP1 antibody (9542, WB 1:500) was from Cell signalling technologies. Rabbit anti-HRS (WB 1:1000) and rabbit anti-ALIX (WB 1:3000) have been described previously[70,71]. Mouse anti-MMP2 (#IM33 clone 42-5D11, WB 1:1000) was from Calbiochem.

Secondary antibodies were from Jackson ImmunoResearch, Molecular Probes, and LI-COR. Oregon Green™ 488 conjugate (G13186), Hoechst 33342 (H3570), Rhodamine-phalloidin (R415), Transferin-Al568 (T23365), EGF-Al488 (E13345) were from Thermo Fisher Scientific/Life Technologies. SAR405 (S7682) was from Sellcckchem. DMSO (D2650), GM6001 (J65687; Alfa Aesar), Batimastat (196440-5MG, Lot 3689083), Marimastat (M2699-5MG, Batch 0000123550), NH₄Cl (1145) were from Merck/Sigma.

### Cell culture

Cell lines were grown according to American Type Culture Collection (ATCC). MDA-MB-231 cells and MCF7 cells were maintained in RPMI

(61870-010 Gibco, Thermo Fisher Scientific/Life Technologies) supplemented with 10% fetal calf serum (F7524; Merck Life Science), 2 mM glutamine (25030–024; Thermo Fisher Scientific), 100 U ml$^{-1}$ penicillin, and 100 µg ml$^{-1}$ streptomycin at 37 °C with 5% $CO_2$. HeLa cells (obtained from Institute Curie, Paris, France) and HT1080 cells were grown in DMEM high Glucose (D0819 from Merck/Sigma) supplemented with 10% fetal calf serum, 100 U ml$^{-1}$ penicillin, 100 µg ml$^{-1}$ streptomycin and maintained at 37 °C with 5% $CO_2$. Cell lines are authenticated by genotyping and regularly tested for mycoplasma contamination.

## Plasmids and generation of stable cell lines

pcDNA-mCh-MT1-MMP and pEGFP-N1-TKS5 were gifts from Philippe Chavrier, Institut Curie, Paris, France. TKS4-GFP was a gift from Arpad Lanyi, University of Debrecen, Hungary[29]. hADAM12-L-myc/his was a gift from Marie Kveiborg, University of Copenhagen, Denmark. ADAM15 (NM_003815) was purchased from Origene as human tagged ORF clone (RC200642) and subcloned as a SNAP-tagged fusion. Truncations, point mutants and fusion proteins were generated with standard molecular biology techniques. MDA-MB-231 cells stably expressing TKS5-GFP, TKS4-mNG, or mCh-MT1-MMP were generated using lentiviral transduction[72]. For all cell lines, a PGK promoter was used, which ensures moderate levels of expression. Third-generation lentiviral particles were generated as follows: GFP-, mNG- or mCherry-fusion proteins were cloned into Gateway pENTR plasmids by conventional restriction enzyme–based cloning or Gibson assembly. From these vectors, lentiviral transfer vectors were generated by recombination into lentiviral Destination vectors derived from pCDH-PGK-MCS-IRES-ResistanceGene (SystemBioSciences) using Gateway LR reactions (Invitrogen). VSV-G pseudotyped lentiviral particles were packaged using a third-generation packaging system (plasmid numbers 12251, 12253, and 12259 from Addgene[73]). Cells were then transduced with virus particles, and stable expressing populations were generated by antibiotic selection. Detailed cloning procedures can be requested from the authors.

## Transient transfections

HeLa cells were transfected with Fugene6 (E2692 from Promega, Lot numbers 0000310309, 0000425615 and 0000466640) according to the manufacturer's instructions. MDA-MB-231 cells were transfected using Lipofectamine 3000 (L3000-015 from Life Technologies/Thermo Fisher Scientific, Lot 2195822) according to the manufacturer's instructions.

## Immunostaining

Cells were seeded on glass coverslips, fixed with 3% formaldehyde (FA; 18814; Polysciences) for 15 min on ice, and permeabilized or not with 0.05% saponin (S7900; Merck/Sigma) in PBS. For cells grown on unlabelled or Oregon Green–labelled gelatin, 0.1% Triton X-100 (Merck/Sigma) was used instead of saponin for 5 min. Fixed cells were then stained with primary antibodies at room temperature for 1 h, washed in PBS/saponin, stained with fluorescently labelled secondary antibody for 1 h, washed in PBS, and mounted with Mowiol containing 2 µg ml$^{-1}$ Hoechst 33342 (H3570; Thermo Fisher Scientific). For the detection of endosomal markers, cells were permeabilized for 5 min on ice with 0.05% saponin in PEM buffer [(0.1 M Pipes (P7643, Merck/Sigma), 2 mM EGTA (E3889, Merck/Sigma), and 1 mM $MgSO_4$ (105886, Merck/Sigma) pH 6.95)] before fixation in order to decrease the fluorescent signal from the cytosolic pool of the proteins[74]. For Airyscan microscopy, nuclei were stained in PBS/Hoechst 33342 (2 µg ml$^{-1}$) for 10 min and mounted in ProLong Diamond (P36961; Thermo Fisher Scientific).

## Confocal fluorescence microscopy, Airyscan microscopy, and image analyses

Confocal micrographs were obtained using an LSM710 or LSM780 confocal microscope (Carl Zeiss) equipped with an Argon-laser multiline (458/488/514 nm), a DPSS-561 10 (561 nm), a continuous-wave laser diode 405–30 CW (405 nm), and a HeNe laser (633 nm). The objective used was a Plan-Apochromat 63×/1.40 oil differential interference contrast (DIC) III (Carl Zeiss). Images were analysed and adjusted (brightness/contrast) in ImageJ/Fiji 1.52p[75] or Zen 2012. For superresolution microscopy, a Zeiss LSM 880 Airyscan (Carl Zeiss) was used with a Zeiss plan-apochromat 63× NA/1.40 oil DIC II objective (Carl Zeiss). The Airyscan detector was either in confocal or super-resolution mode, giving images with voxel size 0.0426 × 0.0426 × 0.1850 µm. Airyscan raw images were processed and aligned in Zen 2012. Images were further processed in ImageJ/Fiji (brightness/contrast[75]) or 3D rendered using Imaris 7.4.2 (Bitplane). All images within one dataset were taken at fixed intensities below saturation, and identical settings were applied for all treatments within one experiment. To analyse transiently transfected cells with comparable expression levels, cells were captured at fixed intensities below saturation, with the criteria that the signal should be clearly visible. Transiently transfected cells which would not meet this criteria were considered having too high or too low expression. In general, at least five (but often more) images were taken randomly throughout the coverslips. Manders colocalisation coefficient was determined with the ImageJ plugin "JACoP"[76]. Specific analyses are described below.

## Automated image analysis of fluorescence intensities and area of gelatin degradation

The NIS-Elements software 5.11 or 5.42.02 was used for background correction ("rolling ball") and automated image analyses. Identical analysis settings were applied for all treatments within one experiment. Intensity based thresholding was used to segment fluorescent dots or areas of gelatin degradation. The total number of cells was quantified by automated detection of Hoechst nuclear stain.

## Fluorescence imaging of cells plated on fibrillar type I collagen

Coverslips were layered with ice cold 2.0 mg ml$^{-1}$ acidic extracted collagen I solution (Corning, 354236, lot 8092003) in 1× MEM (Gibco, 24430-020) mixed with 4% Alexa Fluor 647– conjugated type I collagen. The collagen solution was adjusted to pH 7.5 using 0.34 N NaOH and Hepes was added to 25 µM final concentration. After 3 min of polymerization at 37 °C, the collagen layer was gently washed in PBS before MDA-MB-231-TKS5-GFP-mCh MT1-MMP cells in suspension were added for 4 h at 37 °C. Cells were fixed with 3% formaldehyde (FA; 18814; Polysciences) and mounted for immunofluorescence microscopy.

## Scoring of TKS5 endosome localisation

HeLa cells were transfected with mCherry or mCherry-MT1-MMP wt or ΔC and cotransfected with TKS5 deletion or mutation constructs. Cells were immunolabeled with anti-GFP and anti-mCherry antibody to boost fluorescence and processed for fluorescence microscopy. Quantification was done by visual inspection at a fluorescence microscope: Cells showing expression in the mCherry-channel were subsequently assessed in the green channel and scored for dot-like localisation or diffuse localisation of the TKS constructs. At least 100 cells were counted per condition and experiment and three independent experiments were performed for each construct.

## Live-cell imaging with SAR405

MDA-MB-231-TKS5-GFP-mCh-MT1-MMP cells were grown in MatTek 35 mm glass-bottom dishes (MatTek Corporation) for live-cell imaging on an OMX V4 system (DeltaVision OMX Microscope Applied Precision, GE Healthcare) equipped with an Olympus 60x Plan Apochromat 1.42 numerical aperture objective, three cooled PCO.edge sCMOS cameras, a solid-state light source (InsightSSI) and a laser-based autofocus. Environmental control was provided by a heated stage and an objective heater. 5% $CO_2$ and humidity were

applied using a $CO_2$ mixer (Okolab). One image stack comprising the complete cell volume was acquired every minute for 15 min in total using Softworx software 7.0 (GE Healthcare). After 3 min, DMSO or SAR405 (6 µM endconcentration) was added to the cells.

## Protein purification

Recombinant proteins were expressed as MBP fusion proteins in E. coli. Constructs encoding the PX-domain of TKS5 (amino acids 1-139) as wt, R42, R93 or double mutant were cloned to generate N-terminal MBP-fusion proteins and Rosetta2 (DE3) bacteria were transformed with the resulting plasmids. For protein expression, bacteria were grown in ZYM505 medium[77]. Expression was induced by 0.25 mM IPTG, and induced cells were grown over night at 20 °C. Cells were harvested by centrifugation, resuspended in lysis buffer (50 mM Tris pH7.5, 150 mM NaCl, 1 mM TCEP, cOmplete mini EDTA-free protease inhibitor (Roche)) and lysed by one passage through a homogenizer. Raw lysates were cleared by centrifugation. The MBP fusion proteins were purified by affinity chromatography using amylose resin (NEB E8021). Protein-containing fractions were pooled, dialyzed over night against dialysis buffer (50 mM Tris pH7.5, 150 mM NaCl, 10% Glycerol, 1 mM TCEP). Aliquots were snap-frozen in liquid nitrogen and stored at −80 °C.

## Protein-lipid overlay assays

Protein interaction with lipids was analysed using PIP strips™ (P-6001 Echelon Biosciences, Lot XCM070720-48) following the manufacturer's instructions. In brief, after blocking the membranes in blocking buffer (TBS + 3% fatty-acid free BSA (A7030-100G, Lot SLCD0763)), 0.5 µg ml⁻¹ purified protein was added to the membranes for 1 h at room temperature (RT) with gentle agitation. Following extensive washing of the membranes with TBS-T, bound MBP-fusion proteins were detected with a mouse-anti-MBP antibody (NEB E8032S) followed by goat-anti-mouse-HRP (Jackson 115 035 146) and ECL detection.

## siRNA transfections

HeLa cells were seeded and transfected with siRNA duplexes using Lipofectamine RNAiMax transfection reagent (13778; Thermo Fisher Scientific) following the manufacturer's instructions. For MDA-MB-231 cells, reverse transfection was performed using RNAiMax. For both cell lines, the concentration of individual siRNA duplexes was 10–20 nM. For co-depletion experiments the total concentration of siRNA was 40 nM. Cells were analysed 24–96 h after transfection as indicated in the figure legends. The following human siRNA sequences were used: MT1-MMP/MMP14 (targeting endogenous and exogenous MT1-MMP), 5'-GCA ACA UAA UGA AAU CAC U-3' (s8879, Silencer select siRNA); Clathrin heavy chain: 5'-AUCCAAUUCGAAGACCAAU; HRS: 5'-GCAC-GUCUUUCCAGAAUUC and Silencer Select Negative Controls (4390844 and 4390846) were from Ambion/Thermo Fisher Scientific, RAB7: 5'-CACGUAGGCCUUCAACACAAU (S102662240) was from Qiagen. To target endogenous MT1-MMP/MMP14 a mix of the following three oligos from Qiagen was used: 5'CACAAGGACUUUGCCUCUGA A-3' (SI00071169), 5'CCCUCAGACCUCGCUGGUAAA-3' (SI05042569), 5'GACAGCGGUCUAGGAAUUCAA-3' (SI00071190). AllStars negative control oligo (1027281, Qiagen). TKS4: 5'-GCGAAGACCAAGUCGACAU (J-032834-05), TKS5, 5'-CGA CGG AAC UCC UCC UUU A-3' (J-006657-08) and non-targeting control siRNA (D-001810-01) were purchased from Horizon/Dharmacon.

## Quantitative RT-PCR

Total RNA was extracted using an RNeasy Plus mini kit (74134, Qiagen). cDNA was synthesized using SuperScript IV Reverse Transcriptase (18090010 Thermo Fisher Scientific). Quantitative PCR was performed using the cDNA, SYBR Green I Master Mix (04707516001 Roche Diagnostics), LightCycler 480 (Roche Diagnostics), and QuantiTect Primer Assays (QT00216027 for TKS4 (SH3PXD2B), QT00029764 for TKS5 (SH3MD1), QT00001533 for MT1-MMP (MMP14) and QT00000721 for TATA-binding protein (TBP); Qiagen). Cycling conditions were 5 min at 95 °C followed by 45 cycles for 10 s at 94 °C, 20 s at 58 °C, and 10 s at 72 °C. A standard curve made from serial dilutions of cDNA was used to calculate the relative amount of the different cDNAs in each sample. TKS4, TKS5 and MT1-MMP expression were normalized to the expression of the internal standard TBP.

## Preparation of whole cell lysate

Cells were washed with ice-cold PBS and lysed in sample buffer (125 mM Tris-HCl, pH 6.8, 4% SDS, 20% glycerol, 100 mM DTT, and 0.004% bromophenol blue). To detect CD63 and CD81, DTT was omitted from the sample buffer. Typically, 10-15% of a confluent 6-well dish was loaded per lane.

## Immunoblotting and quantification of Western blots

Samples were subjected to SDS-PAGE on 10% (567−1034; Bio-Rad) or 4−20% (567−1094; Bio-Rad) gradient gels and blotted onto Immobilon-P membranes (IPVH00010; Merck Millipore). Membranes incubated with fluorescently labelled secondary antibodies (IRDye680 and IRDye800; LI-COR) were developed by Odyssey 3.0.30 infrared scanner (LI-COR) or Azure Sapphire Biomolecular Imager. Membranes detected with HRP-labelled secondary antibodies were developed using Clarity Western ECL substrate solutions (Bio-Rad) with a ChemiDoc XRS+ imaging system (Bio-Rad) or Azure Sapphire Biomolecular Imager. Quantification of WB data was performed with ImageJ/Fiji 1.52p, Odyssey 3.0.30 or Azure Spot 2.1.097 (Azure Biosystems) and samples were normalized to a loading control (Actin, GAPDH or Vinculin). Uncropped Western blot images are displayed in Supplementary Fig. 13.

## Co-Immunoprecipitations

For immunoprecipitation experiments, cells at 70−80% confluency in one 10 cm dish per condition were washed twice with ice-cold PBS before lysis in 25 mM HEPES, 125 mM potassium acetate, 2.5 mM magnesium acetate, 5 mM EGTA, pH 7.2, freshly supplemented with 0.5% IGEPAL, 1 mM dithiothreitol (DTT), protease inhibitor cocktail (Merck/Sigma) and PhosSTOP™ (Merck/Sigma). Lysates were centrifuged for 5 min at 18,800 × g at 4 °C and supernatants (SN) were immunoprecipitated with the indicated antibodies and Dynabeads™ Protein G (10004D, Thermo Fisher Scientific) rotating for 60 min at 4 °C. The immunoprecipitates were washed three times in lysis buffer, eluted with 2x sample buffer and subjected to immunoblotting as described above. Typically, 1.5−2% of cell lysate and 50% of the washed IP samples were used for Western blotting analysis.

## MBP pulldown assay

For MBP pulldown assays, cells were washed twice with icecold PBS before lysis in 25 mM HEPES (pH 7.2), 125 mM potassium acetate, 2.5 mM magnesium acetate, 5 mM EGTA, pH 7.2, freshly supplemented with 0.5% IGEPAL, 1 mM dithiothreitol (DTT), protease inhibitor cocktail (Merck/Sigma) and PhosSTOP™ (Merck/Sigma). Lysates were centrifuged for 5 min at 18,800 × g at 4 °C. SN was mixed with 10 µg purified protein and 25 µl amylose resin (E8021L, New England Biolabs) and incubated for 60 min rotating at 4 °C. After washing three times in lysis buffer, the SN was completely removed with a hollow needle and 2x sample buffer was added to each sample. Samples were boiled and subjected to immunoblotting as described above.

## Extracellular vesicle (EV) fraction enrichment

EVs were isolated as previously described[78]. Briefly, cells were seeded at a density of $3 \times 10^6$ cells per 150 mm dish. After 3 days, the cells were washed twice with 12 mL serum-free medium (SFM) and incubated for 36 h and with 14 ml of SFM. The conditioned medium (CM) was

collected and centrifuged at $300 \times g$ for 10 min at 4 °C to remove cells and at $1000 \times g$ for 10 min at 4 °C to remove large cellular debris. The pellet is the apoptotic body enriched "cell debris fraction". Thereafter the SN was centrifuged at $10,000 \times g$ at 4 °C for 30 min and then ultracentrifuged at $100,000 \times g$ at 4 °C for 70 min. The pellet was washed with PBS, and centrifuged again at $100,000 \times g$ for at 4 °C for 70 min. The pellet (= EV fraction) was resuspended in 2x sample buffer and 30% of the EV fraction was used for Western blotting per lane. EVs isolated by this method have been characterized previously using electron microscopy and Nanoparticle Tracking Analysis[79]. The cells were detached using trypsin/EDTA and the cell numbers were counted. Cell lysate from $5 \times 10^4$ cells were used for WCL Western blotting per lane.

## Correlative light and electron microscopy (CLEM)
MDA-MB-231-TKS5-GFP-mCh-MT1-MMP cells grown in CLEM imaging dishes were fixed with 3% PFA in 0.1 M PHEM buffer at RT for 10 min and stained with Hoechst 33342. Confocal imaging was done using a Zeiss LSM880, followed by fixation with 1% GA in 0.1 M PHEM buffer for 30 min at RT and over night at 4 °C. The cells were postfixed and contrasted in 1% osmium tetroxide, 0.25% tannic acid and 4% uranyl acetate before stepwise dehydration to 100% ethanol and flat-embedding in Epon. Serial sections (200 nm) were cut on an Ultracut UCT ultramicrotome (Leica) and collected on formvar-coated slot grids. The sections were observed at 200 kV with a Thermo Scientific Talos F200C microscope with a Ceta 16 M camera.

## Immuno-electron microscopy (Tokayasu)
Cells were fixed using 4% methanol-free paraformaldehyde and 0.1 glutaraldehyde (Electron Microscopy Sciences, Hatfield, PA) in 0.2 M HEPES and prepared for cryo immuno-electron microscopy basically as described in ref. 80. Labelling for mCherry was done using goat anti mCherry followed by rabbit anti goat IgG (Cappel Research Reagents, ICN Biochemicals, Irvin, CA) and finally protein A coated colloidal gold (G. Posthuma, Utrecht, The Netherlands). Sections were examined using a Tecnai G2 Spirit TEM (FEI, Eindhoven, The Netherlands) equipped with a Morada digital camera using iTEM (SIS) software (Olympus Soft Imaging Solutions, *Münster*, Germany). Images were processed using Adobe Photoshop.

## Immuno-electron microscopy (silver enhancement)
Cells cultured on cover slips were fixed with 4% PFA in 0.1 M PHEM buffer for 15 min and permeabilized with 0.05% saponin in 1x PBS prior to immunolabeling with primary antibodies anti-GFP (ab6556 from Abcam, 1:500) or anti-MT1-MMP (MAB3328 from Merck/Sigma, 1:100) for 1 h at RT and secondary antibodies (Invitrogen Molecular Probes A24923 or A24921, Alexa Fluor 594 Nanogold, 1:200) for 45 min at RT. The cells were further fixed with 1% GA in 0.1 M PHEM buffer for 30 min at RT and over night at 4 °C before silver enhancement according to the manufacturer´s protocol (HQ silver enhancement kit, Nanoprobes) and gold toning. Postfixation, embedding, sectioning and imaging were done as described for CLEM.

## Immuno-electron microscopy characterisation of EVs
EVs were isolated by $100,000 \times g$ ultracentrifugation of conditioned medium of MDA-MB-231-TKS5-GFP-mCh-MT1-MMP cells, as described above. The isolated EVs were added to glow-discharged charged EM grids and immunolabeled as follows: Blocking in 0.1 M PHEM with 0.5% BSA for 5 min, primary antibody (mouse anti-CD63, clone H5C6 DHB, 1:50 or goat anti-mCherry, AB0040-200 from Acris, 1:50) for 15 min, secondary antibody (rabbit anti-mouse, 110-4102 from Rockland, 1:1500, or rabbit anti-goat, P044901-2 from Dako, 1:250) for 15 min, and protein A gold 10 nm (CMC, Utrecht, Netherlands, 1:50) for 15 min. Further, the samples were fixed using 1% GA in 0.1 M PHEM before contrasting with 4% uranyl acetate in $H_2O$. After drying, the samples

were observed either with a JEOL-JEM 1230 electron microscope at 80 kV with a Morada camera (Olympus), or with a Thermo Scientific Talos F200C microscope at 200 kV with a Ceta 16 M camera.

To obtain correct measure of the diameter of sphere shaped EVs, the measured diameter of the collapsed EVs was adjusted using the following formula:

$$R = \frac{\sqrt{\pi(r_1^2 - r_0^2) + 2r_0^2}}{2} \tag{1}$$

## Collecting and concentration of conditioned medium
$3 \times 10^6$ MDA-MB-231 cells and MDA-MB-231-TKS5-GFP-mCh-MT1-MMP cells, or $1.6 \times 10^6$ HT1080 cells in 150 mm dishes were transfected with different siRNAs. After 3 days, the cells were washed twice with 12 mL SFM and incubated for 24 h in 14 ml of SFM. The conditioned medium was collected and centrifuged at $300 \times g$ at 4 °C for 10 min, at $1000 \times g$ at 4 °C for 10 min and at $10,000 \times g$ at 4 °C for 30 min. Then the conditioned medium was concentrated to 300–400 μl by an AmiconUltra-15 centrifugal filter unit (10 kDa cutoff; UFC901024 Millipore/Merck/Sigma) at $5000 \times g$ for 30 to 40 min at 4 °C. The cells were detached using trypsin/EDTA and the cell numbers were counted. Cell lysate from $1.5 \times 10^5$ cells was used for WCL Western blotting per lane. The volume of concentrated conditioned medium (CM) used in experiments was normalized by the cell numbers. 5% of CM from MDA-MB-231-TKS5-GFP-mCh-MT1-MMP cells or 10% from MDA-MB-231- or HT1080 cells was used for Western blotting analysis. The remainder was used to treat recipient cells.

## Coating of coverslips with gelatin
Oregon Green™ 488–conjugated gelatin-coated (G13186, Thermo Fisher Scientific) coverslips were prepared as previously described[81]. In short, coverslips (12 mm diameter, No. 1 thickness; 631-0666, VWR International) were precleaned in 4% nitric acid for 30 min. After washing, the coverslips were coated with 50 μg ml$^{-1}$ poly-L-lysine (P7890, Merck/Sigma) for 30 min, washed in PBS, and fixed with cold 0.5% glutaraldehyde (G5882, Merck/Sigma) in PBS for 15 min on ice. Subsequently, the coverslips were washed in PBS and coated for 10 min with preheated (44 °C) 10 mg ml$^{-1}$ unlabelled or Oregon Green™ 488–conjugated gelatin/2% sucrose in PBS. After coating, the coverslips were washed with PBS and incubated in 5 mg ml$^{-1}$ sodium borohydride (452882, Merck/Sigma) for 15 min. The coverslips were then washed with PBS, sterilized with 70% ethanol, and equilibrated in serum-containing medium for 1 h at 37 °C before addition of cells.

## Gelatin degradation assay
Hela cells were seeded on coverslips coated with Oregon Green™ 488-conjugated gelatin (G13186, Thermo Fisher Scientific). After 3 h the concentrated CM was supplemented with 1% pen/strep and 10% FBS, split into 2 equal parts and added into duplicated wells. After 24 h cells were fixed in 3% FA in PBS for 15 min, permeabilized with 0.1% Triton X-100 in PBS, incubated with Rhodamine-phalloidin and Hoechst 33342 for 15 min, and mounted for examination by confocal microscopy. For experiments using siRNA-mediated depletion of MT1-MMP, or TKS4/5, conditioned medium was collected as described in "Collecting and concentration of conditioned medium". 96 h after siRNA transfection the knockdown efficiencies were checked by Western blotting and measurements of gelatin degradation were performed. Samples were analysed using a LSM710 confocal microscope (Carl Zeiss); a 63× objective and zoom 1.0. Cells/fields of imaging were chosen on basis of the nuclear staining, and at least 20 images were randomly taken throughout the coverslips in each experiment, with typically 4–5 cells per image. All images within one experiment were taken with constant gain and pinhole parameters. Area of gelatin

degradation was measured automatically by NIS-Elements 5.11 or 5.42.02 and related to the total number of cells.

## Live-cell imaging in 3D type I collagen

Live cell imaging in 3D type I collagen was essentially done as described in[82]. In short, 8 well glass bottom slides (IBIDI, 80827) were layered with 5 mg ml$^{-1}$ unlabelled type I collagen (Corning, 354249, lot 9161007) mixed with 1/25 volume of Alexa Fluor 647–labelled collagen and polymerized for 3 min at 37 °C before the bottom layer of collagen was gently washed in PBS. 150 μl of cell suspension (7.5×10^3 cells/well) in full medium was added and incubated for 30 min at 37 °C. Medium was gently removed, and two drops of a mix of Alexa Fluor 647–labelled type I collagen/unlabelled type I collagen was added to a final concentration of 2.0 mg ml$^{-1}$ (Corning, 354236, lot 8092003) on top of the cells. After polymerization for 90 min at 37 °C 150 μl of either regular full medium or conditioned medium from MDA-MB-231-TKS5-GFP-mCh-MT1-MMP (CM) cells was added, and time-lapse imaging was started 60 min after. Z-stacks of images were acquired every 20 min for up to 24 h on a Nikon ECLIPSE Ti2-E inverted microscope (Nikon Corp, Tokyo, Japan) equipped with a CSU-W1 dual spinning disc (50 um pinholes & 50 um pinholes with microlenses) confocal unit (Yokogawa Electric Corp, Tokyo, Japan), a Prime BSI sCMOS camera (Teledyne Photometrics, Tucson, AZ, US), a laser unit with 405/488/561/638 nm lasers (120/100/100/100 mW), and BrightLine single-band bandpass filters (447/60 nm, 525/50 nm, 600/52 nm, 708/75 nm). The objective used was CFI Plan Apo λ 40x (NA 0.95, Air). Images of 5–8 random fields of view in each well were captured. Images were acquired and processed using NIS-Elements-AR software (version 5.30.04, denoise, max projections) before single cells were manually tracked by using ImageJ/Fiji 1.52p, and further analysed using Chemotaxis and Migration Tool 2.0 (IBIDI GmbH). To eliminate systematic differences between experiments, data were normalized by the mean of all conditions and experiments. Images shown are maximum intensity projections of Z-sections and were adjusted by linear brightness-contrast adjustments.

## Spheroid assay

HeLa cells were seeded (1×10$^3$ cells per well) in 96-well ultralow-attachment plates from Merck Life Sciences (CLS7007, Merck/Sigma) and allowed to form multicellular spheroids for 24 h. Then 2/3 of the medium was removed and type collagen I (Corning, 354236, lot 8092003) was gently added to a final concentration of 2 mg ml$^{-1}$ and allowed to polymerize for 1 h at 37 °C. Spheroids were then imaged by OlympusIX81 microscope, using a positive-low (PL) 4x phase contrast objective (day 0) before regular full medium or conditioned medium from MDA-MB-231-TKS5-GFP-mCh-MT1-MMP was added. Plates were placed in the incubator at 37 °C for 7 days and the spheroids were inspected and imaged every 24 h. On day 3 regular full medium and conditioned medium from MDA-MB-231-TKS5-GFP-mCh-MT1-MMP were again added to the spheroids. Quantification of individual spheroid area was done using ImageJ/Fiji 1.52p. The area of the whole spheroid (day 3 and 7) was divided by the area of the dense core of the spheroid (day 0) to get fold increase.

## Cell migration

HeLa cells were seeded in 96-well plates (4 × 10$^3$ cells per well) and allowed to attach. In the meantime, conditioned medium from MDA-MB-231-TKS5-GFP-mCh-MT1-MMP cells was harvested and concentrated as described above. The Incucyte® Nuclight Rapid Red Dye for Live-Cell Nuclear Labelling (Item No. 4717, Sartorius) was added to the concentrated CM or control culture medium before addition to the cells. Live cell fluorescence imaging was performed using an Incucyte S3 (2021 C) live-cell microscope, equipped with a 4x objective at 15 min intervals for 24 h. Random cell migration was measured using the NIKON NIS-Elements software 5.42.02: In brief,

red fluorescent nuclei were segmented using "segment.ai". Automated tracking was performed using the tracking module in NIS-Elements.

## Inverted invasion in collagen-I

For inverted invasion assays we used transwell filter inserts with 8 μm pores (Costar 3422), filled with a collagen-I matrix prepared as follows: Rat tail collagen I (Corning 354236) was diluted to 2.2 mg/ml using sterile water and mixed with 10× MEM (Gibco 21430020) and HEPES (Gibco 21430020, 25 mM endconcentration), everything on ice. The collagen solution was neutralized with NaOH to reach a pH of 7.5 and a final concentration of 2 mg/ml before addition to the transwell inserts. Polymerization was allowed in a 37 °C incubator for 30–45 min. The inserts were then inverted, and 5 × 10$^4$ HeLa cells were seeded on top of the filter on the opposite side of the collagen matrix. Cells were allowed to adhere to the filter for 3–4 h. In the meantime, non-invasive HeLa cells or invasive MDA-MB-231-TKS5-GFP-mCh-MT1-MMP cells were seeded in serum free medium. The filters with the attached HeLa cells were washed three times in serum free medium before the inserts were turned and placed in the wells containing invasive or non-invasive cells for co-culture. The upper chamber was filled with serum-containing medium and HGF (100 ng/ml) serving as chemoattractants. During the 96 h invasion period the inserts were moved twice to fresh co-culture cells in SFM to avoid cell death. Finally, cells were stained with Calcein AM (C3100MP, Thermo Fisher Scientific, 4 μM endconcentration) for 1 h before invading cells were visualized by spinning disc microscopy (Nikon CREST microscopy, 20x air objective, 5 μm z-stack intervals). Sections of 5 μm intervals were captured at 5–6 random locations for each plug, 3 plugs were made for each condition in each of three independent experiments. Images were analysed with Nikon NIS elements using cell segmentation based on the Calcein AM staining and counting of the number of cells that invaded further than 40 μm into the collagen matrix.

## Gelatin zymography

Concentrated conditioned medium (CM) was mixed with an equal amount of Novex Tris-Glycine SDS Sample Buffer (2X) (LC2671, Thermo Fisher Scientific) without heating and then subjected to SDS-PAGE on Novex 10% Zymogram Plus (Gelatin) gels (ZY00100BOX, Thermo Fisher Scientific) with a constant voltage of 125 V for 135 min. After electrophoresis, the gels were incubated in Zymogram Renaturing Buffer (LC2670, Thermo Fisher Scientific) for 30 min with agitation to remove SDS. Then the gels were equilibrated in Zymogram Developing Buffer (LC2671, Thermo Fisher Scientific) for 30 min with agitation followed by incubation overnight at 37 °C in fresh Zymogram Developing Buffer. The gels were subsequently stained with Colloidal Blue Staining Kit (LC6025, Thermo Fisher Scientific) and washed with deionized water for at least 2 h. Areas of gelatin proteinase activity appeared as clear bands against a blue background which were imaged using the Odyssey 3.0.30 (LI-COR) infrared scanner.

## Protein structure prediction

The PX domain (amino acid 1-130) of human TKS4 and TKS5 was predicted with ColabFold using the algorithms of AlphaFold2 and MMseqs2 for sequence alignments[83,84]. For each protein, five models were predicted and the models with the highest confidence were selected. The ribbon model representations were made with ChimeraX[85].

## Statistical analysis and considerations

The number of individual experiments and the number of cells or images analysed are indicated in the figure legends. The number of experiments was adapted to the expected effect size and the anticipated consistency between experiments. We tested our

datasets for normal distribution by Kolmogorov–Smirnov, D'Agostino and Pearson, and Shapiro–Wilk normality tests, using GraphPad Prism Version 8.02. For parametric data, an unpaired two-sided $t$ test was used to test two samples with equal variance, and a one-sample $t$ test was used in the cases where the value of the control sample was set to 1. For more than two samples, we used one-way ANOVA with a suitable post hoc test. For nonparametric samples, Mann–Whitney test was used to test two samples and Kruskal–Wallis with Dunn's post hoc test for more than two samples. All error bars denote mean values ± SD or SEM, as indicated in every figure legend ($*P < 0.05$; $**P < 0.01$; $***P < 0.001$; $****P < 0.0001$, ns not statistically significant).

### Reporting summary

Further information on research design is available in the Nature Portfolio Reporting Summary linked to this article.

### Data availability

All data shown and used to generate plots, as well as detailed statistical information accompanies this manuscript in the source data file. Uncropped western blots are shown in the supplementary information. Underlying image data are available from the corresponding author upon request. Source data are provided with this paper.

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

## Acknowledgements

We thank Anne Engen for expert help with cell cultures. The Core Facilities for Advanced Light Microscopy and Advanced Electron Microscopy at Oslo University Hospital are acknowledged for providing access to and training on relevant microscopes. We thank Ulrikke Dahl Brinch for excellent technical help with sample preparation for electron microscopy. Felix Margadant contributed to the calculation of EV diameter. EMW is a research fellow of InvaCell (donation from Mr. Trond Paulsen). CR is supported by the Norwegian Cancer Society (project number 198140 and 246670). HS is supported by grants from the Research Council of Norway (project number 302994), the South-Eastern Norway Regional Health Authority (project number 2018081), The Norwegian Cancer Society (project number 182698), and the European Research Council (project number 788954). This work was partly supported by Norges Forskningsråd through its Centres of Excellence funding scheme (project number 262652). Figures were created using Adobe Illustrator CS6 and BioRender (https://biorender.com/).

## Author contributions

EMW and NMP contributed to conceptualization, investigation, validation, formal analysis, visualization and reviewing and editing. LAE, LW, IK and ES contributed to investigation, validation, visualization and reviewing and editing. LM contributed with investigation, reviewing and editing. AB contributed with validation, reviewing and editing. HS contributed with funding acquisition, resources, supervision, and reviewing and editing. CR contributed with conceptualization, investigation, validation, formal analysis, visualization, funding acquisition, supervision, project administration and reviewing and editing. EMW and CR wrote the original draft. All coauthors have seen the final draft.

## Competing interests

The authors declare no competing interests.
