## [Peer Review File · Nature Communications]

Intercellular transfer of cancer cell invasiveness via endosome-mediated protease sheddingREVIEWER COMMENTS

Reviewer #1 (Remarks to the Author):

Transmembrane matrix metalloproteinase MT1-MMP/MMP14 promotes cancer cell invasion and stimulate tumor metastasis in vivo. However, the mechanism through which MT1-MMP/MMP14 stimulate tumor invasive is yet to be elucidated. Here, Eva Maria Wenzel et al. demonstrated TKS4/5 linked MT1-MMP/MMP14 to ADAM and cleaved into ectodomain. Afterwards, these ectodomain was transferred to surrounding non-invasive cells through CM or secreted EVs and turned non-invasive cell to invasive cell. This paper elucidated TKS4/5 regulated MT1-MMP/MMP14 transfer between invasive cancer cell to non-invasive cell and turned the later one invasive. This manuscript will no doubt be of interest to the wider research and clinical community, yet there are aspects that could be strengthened to support hypotheses throughout. The main points are:

- 1) TKS4/5 was highly recruited to endosome under MT1-MMP overexpression situation, but the expression level in protein or RNA level of TKS4//5 and MT1-MMP was not determined in different transfected cells. So, author should test and keep comparable amount of MT1-MMP or TKS4/5 in different transfected cells. In addition, a scramble protein, not but only m-Cherry, should be used as the TKS4/5 control in Figure1b/c/d.
- 2) Does MT1-MMP over expression impact the production of EVs? Besides, what is the characterization of these EVs? Are they exosomes or microparticles? Authors need to check the diameter, and some other markers, e.g CD81, Alix and CD63 etc, need to be detected as well.
- 3) Author demonstrated ADAM15 preferred to interact with MT1-MMP and TKS4 by immunoprecipitation in Figure 4i, but this data is not solid enough. Specifically, the ADAM-15-myc-SNAP in WT lane of IP (anti-mCh) is comparable or subtle difference with other two lanes, author have to repeat this experiment and get more convincible data.
- 4) GM6001, which mainly inhibits MT1-MMP activity, only has marginal effect on 34 kDa band decrease, why? And why ADAM15- myc-SNAP was preferentially found in a complex with mCh-MT1-MMP in the presence of TKS4 but not TKS5?
- 5) The MDA-MB-231 cell derived CM transferred the invasiveness to non-invasive cells, but whether the CM changed intrinsic MT1-MMP or TKS4/5 expression?
- 6) Author should employ more methods and provide more evidence to demonstrate CM changed the invasiveness of recipient cells besides testing the distance in collagen-I. For example, author could conduct trans-well, wound healing etc. and check some genes or proteins involved in cell movement.
- 7) The whole study was completely performed in vitro, despite these current data seems to be solid, the body context will be much more complicated and different compared with in vitro scenario. Author should test TKS4/5 and MP1-MMP mediated intercellular invasiveness transfer happens in vivo as well, of course, this would be incredible hard since the different molecules tracking in vivo is challengeable.

However, author should at least determine the impact of TKS4/5 or MT1-MMP depletion on MDA-MB-231 cell invasiveness in vivo and on matrix extracellular matrix degradation.

8) To demonstrate MDA-MB-231 derived CM or EVs drive the non-invasive cancer cell to be invasive, author need to inoculate non-invasive cells in mice followed by exogenous CM/EVs injection, then evaluate if invasiveness would be changed.

9) Does this MT1-MMP+TKS4/5 mediated intercellular invasiveness also suite for other types of cancer?

10) Do cancer cells from metastatic tumor express higher TKS4/5 compared to one from non-metastatic tumor? Is there correlation between TKS4/5 and survival of patients with cancer?

Reviewer #2 (Remarks to the Author):

This paper from the Raiborg lab describes experiments characterising mechanisms responsible for influencing intracellular trafficking and proteolytic cleavage of the transmembrane matrix metalloprotease, MT1-MMP, and how shedding of MT1-MMP's proteolytic cleavage products can impart invasive characteristics to other cells that do not express MT1-MMP. The authors characterise details of an association formed between MT1-MMP's cytosolic domain and the invadopodia marker proteins TKS4&5, how this depends on TKS4&5's PX domains and how this promotes recruitment of MT1-MMP to the intraluminal vesicles of late endosomes. These protein:protein interactions and their ensuing trafficking to the low pH environment of the late endosome lumen allow ADAM-mediated cleavage of MT1-MMP. The authors then proceed to show that MT1-MMP released from cells – in a manner that is dependent on TKS4&5 – can encourage other cells that do not express MT1-MMP to degrade collagen and become invasive themselves.

In general, the work described in this paper is of great interest, well-executed and convincing and fully support the conclusions drawn by the authors. In particular, the cellular mechanisms influencing how MT1-MMP:TKS4/5 interaction influences MT1-MMP trafficking and how these trafficking events, in turn, can foster the appropriate conditions to allow ADAM-mediated MT1-MMP cleavage are carefully and convincingly carried-out. The experiments demonstrating that conditioned medium from cells that are competent to execute TKS-dependent trafficking, and thus ADAM-mediated cleavage of MT1-MMP, can encourage HeLa cells to become invasive are also very interesting.

However, I feel that more clarification is needed to justify publication of this paper in Nature Comms. I would highlight my major points 1 and 2 below as being most important. Specifically, that the authors determine:

a) the extent to which the interesting phenomena they describe depend on overexpression of MT1-MMP and;

b) the role of intraluminal budding and EV production, as opposed to just sorting of MT1-MMP to MVBs/late endosomes, in cleavage of MT1-MMP and intercellular transfer of cancer cell invasiveness.

Major points:

1. Most of the experiments are conducted in cells that overexpress an mCherry-tagged MT1-MMP construct. Therefore, the authors should conduct experiments to address the following two questions:

(i) Can TKS-dependent trafficking and cleavage of MT1-MMP be demonstrated for the endogenous transmembrane MMP? This could be determined in cancer cells, in addition to MDA-MB-231s, that endogenously express high levels of MT1-MMP

(ii) Does the inclusion of the mCherry tag in the juxtamembrane ectodomain influence MT1-MMP's TKS-dependent trafficking and cleavage?

2. The authors show that MT1-MMP and some of its cleavage fragments are released from cells both in association with EVs and in solution. They also demonstrate in Fig. 6e that both EV-associated and soluble MT1-MMP (fragments) can transfer invasiveness to HeLa cells. In these regards, can the authors address the following questions:

(i) To what extent are ILV-budding and EV production/release actually required for TKS-dependent cleavage of MT1-MMP and release of MT1-MMP (fragments) capable of intercellular transfer of invasiveness?

(ii) How does interfering with intraluminal budding, for instance by KD/KO of CD63, Tsg101 and/or ESCRTs affect MT1-MMP cleavage/release? The work would also be strengthened if the authors included markers such as CD63, Alix or ESCRT-related proteins to show endocytic origin in their EV preps (note: the CD9 antibody they are using has been discontinued due to low specificity and the blots are very faint for CD9 in WCL, for instance in figure 3c).

(iii) What is the effect of knockdown of Rab27s, or other ways to oppose LE/MVB docking/fusion (such as VAMP7 knockdown) on release of MT1-MMP fragments and intercellular transfer of invasiveness?

3. The authors have shown very nicely that non-MT1-MMP1-expressing recipient cells (HeLas) are required for released MT1-MMP fragments to promote invasion. But what happens to soluble and EV-associated MT1-MMP fragments when they encounter the HeLa cells? Do they become associated with domains on the HeLa cell surface? Are they internalised by HeLas and, if so, is this required for the HeLas to enact collagen degradation/invasion?

4. What is the topology of MT1-MMP and TKSs in EVs? Can the authors determine this using immunogold EM?

5 . I'm not sure I agree that Marimastat is more potent than GM6001 in generating the 34kDa fragment of MT1-MMP. From the blot in Fig. 4g it looks very similar to GM6001.

6. I don't think that the data presented indicate that the TKs PX domain alone is sufficient for MT1-MMP-dependent endosomal localisation. From Fig. 2b it looks as if TKs 1-139 is recruited to endosomes in the presence of MT1-MMP whether it possesses the (PX-binding) cytotail...so this looks non-specific to me. Better to say that the PX domain is necessary, but not necessarily sufficient, for endosomal localisation of TKS5.

Minor comments:

1. It would be helpful if the authors could indicate, in addition to the constructs used for stable expression in Fig. 2a, the siRNA targeting the endogenous MT1-MMP (as stated in the main text and respective figure legend).

2. In figure 6F the authors should justify the use of the term "exosomes" instead of EVs when explaining the schematic model of this figure.

Response to reviewers:

We thank the reviewers for their very insightful and helpful comments, which have guided us to strengthen our conclusions and improve the quality of our manuscript. Our replies to the reviewers are included below in blue font, with original reviewer comments in black.

Reviewer #1 (Remarks to the Author):

Transmembrane matrix metalloproteinase MT1-MMP/MMP14 promotes cancer cell invasion and stimulate tumor metastasis in vivo. However, the mechanism through which MT1-MMP/MMP14 stimulate tumor invasive is yet to be elucidated. Here, Eva Maria Wenzel et al. demonstrated TKS4/5 linked MT1-MMP/MMP14 to ADAM and cleaved into ectodomain. Afterwards, these ectodomain was transferred to surrounding non-invasive cells through CM or secreted EVs and turned non-invasive cell to invasive cell. This paper elucidated TKS4/5 regulated MT1-MMP/MMP14 transfer between invasive cancer cell to non-invasive cell and turned the later one invasive. This manuscript will no doubt be of interest to the wider research and clinical community, yet there are aspects that could be strengthened to support hypotheses throughout. The main points are:

1) TKS4/5 was highly recruited to endosome under MT1-MMP overexpression situation, but the expression level in protein or RNA level of TKS4//5 and MT1-MMP was not determined in different transfected cells. So, author should test and keep comparable amount of MT1-MMP or TKS4/5 in different transfected cells. In addition, a scramble protein, not but only m-Cherry, should be used as the TKS4/5 control in Figure1b/c/d.

We agree with the reviewer that the expression level of the transiently overexpressed proteins should be within a similar range in order to qualitatively compare MT1-MMP-dependent endosomal recruitment of TKS4/5, and that relevant controls should be included. We have taken several measures to ensure this:

First, images were captured with identical settings, below pixel value saturation, with the criteria that the signals should be clearly visible. The presented images are identically adjusted. This allowed us to analyze transiently transfected cells with comparable expression levels between the conditions. We apologize that this was not explained properly in the manuscript, and we have now added a clarification in the methods section on page 18-19, lines 592-596.

Second, we have verified the average protein expression of transiently expressed TKS4/5 wt and mutation constructs by Western blotting, which showed uniform levels of overexpression for all TKS4/5 constructs (old Fig. S3a,c, new Supplementary Fig. 2a,c).

Third, and importantly, the mCh-MT1-MMP-dependent endosomal recruitment of TKS4/5-GFP observed by transient transfection in Fig. 1 was confirmed quantitatively using cells stably expressing these proteins (old Fig. 2a, new Fig. 1e), and their expression level was characterized by Western blotting in old Fig.S2a (new Supplementary Fig. 1c) and throughout the manuscript. Thus, the images in Fig. 1 serve as an introduction and an initial qualitative observation that we verify quantitatively and study in depth throughout the manuscript using stable cell lines.

Fourth, to strengthen our conclusion regarding MT1-MMP-dependent recruitment of TKS4/5 to endosomes we use different relevant controls in our study. In Fig. 1b, we directly compare the recruitment of endogenous TKS5 to endosomes in mCh-MT1-MMP overexpressing cells vs. non-transfected neighbouring cells. We believe that this direct comparison between cells in the same field of view strengthens the conclusion that endogenous TKS5 is recruited to endosomes upon MT1-MMP overexpression, a phenomenon that we continue to study in depth in the manuscript by different approaches. In the transient transfection approach in Fig. 1c, d, we used mCherry as a negative control. We agree with the reviewer that mCherry should not be the only control. Instead of introducing the

transient expression of a scrambled protein, as the reviewer suggests, we continued our study including the C-terminally deleted mCh-MT1-MMP (Δ C). In both transient and stable cell systems, this endosomally localizing deletion mutant of mCh-MT1-MMP failed to recruit TKS4/5 and thus behaved as a proper negative control in our experiments, when compared to the full-length version of mCh-MT1-MMP (old Figs. 2a, S2). To introduce this important control early in the manuscript, we have now moved these figures to new Fig. 1e, new Supplementary Fig. 1. Importantly, the expression levels of mCh-MT1-MMP WT and Δ C are comparable as judged by WB (old Fig. S2a, new Supplementary Fig. 1c).

2) Does MT1-MMP over expression impact the production of EVs? Besides, what is the characterization of these EVs? Are they exosomes or microparticles? Authors need to check the diameter, and some other markers, e.g CD81, Alix and CD63 etc, need to be detected as well. In the revised manuscript, we characterize the EV fraction by Western blot analyses with the following EV markers, as suggested: Alix, CD9, TSG101, CD81 and CD63 (new Fig. 3c, new Fig. 8a, new Supplementary Fig. 12f). According to this characterization, overexpression of MT1-MMP does not seem to change the production of EVs. Western blot analysis of parental and mCh-MT1-MMP overexpressing cells shows that the EV fraction contains equal amounts of the EV markers TSG101, CD9 and Syntenin-1, a direct interactor of Alix (new Fig. 3c).

As suggested, we have also characterized the EVs by electron microscopy, and find that the majority have a diameter between 40 and 100 nm, consistent with the size of intraluminal vesicles (ILVs) of endosomes. Immuno-gold labelling of the EVs demonstrates that the 40-80 nm EVs have the highest density of CD63 labelling (new Fig. 3d). Thus, we conclude that the majority of the EVs are exosomes. This notion is further strengthened by functional experiments where we abrogate ILV formation and/or exosome secretion and observe a reduced transfer of invasiveness (new Fig. 8).

3) Author demonstrated ADAM15 preferred to interact with MT1-MMP and TKS4 by immunoprecipitation in Figure 4i, but this data is not solid enough. Specifically, the ADAM-15-myc-SNAP in WT lane of IP (anti-mCh) is comparable or subtle difference with other two lanes, author have to repeat this experiment and get more convincible data.

We apologize that the quality of the presented Western blot was not adequate. We have now added an improved version, where we have used the more sensitive method chemiluminescence instead of fluorescence for detection. Importantly, the Co-IP data are supported by quantifications from three independent experiments, which show that ADAM15 is more associated with MT1-MMP in the presence of TKS4 wt, but not when TKS4 is mutated and unable to interact with MT1-MMP (new Fig. 5c, new Supplementary Fig. 8c).

Furthermore, we have performed additional co-immunoprecipitation experiments and quantifications to strengthen the finding that ADAM15 is more associated with MT1-MMP in the presence of TKS proteins: In addition to the previous TKS4-data, we now show that ADAM-15-myc-SNAP is preferentially found in complex with mCh-MT1-MMP in the presence of TKS5-GFP and in cells co-expressing TKS4-GFP and TKS5-GFP. In all cases, mutated TKS4/5-GFP, unable to interact with MT1-MMP, fails to recruit ADAM15 to the complex (new Fig. 5c, new Supplementary Fig. 8c).

We observed that only a minor fraction of cellular ADAM15 is found in a ternary complex with MT1-MMP and TKS4/5 proteins. TKS5 has been shown to interact with several of the ADAM family of sheddases, which might be present in the complex at the same time. Being enzymes, only small amounts of ADAM-recruitment by TKS4/5 would be sufficient to mediate cleavage of MT1-MMP. This notion is supported by other data in the manuscript, showing that the cleavage of MT1-MMP is sensitive to Batimastat, and depending on TKS4/5.

4) GM6001, which mainly inhibits MT1-MMP activity, only has marginal effect on 34 kDa band decrease, why?

The reviewer is right, that GM6001 mainly inhibits MT1-MMP activity. The marginal effect of GM6001 is consistent with the 34 kDa band of mCh-MT1-MMP being a product of non-autocatalytic cleavage. Being a metalloprotease inhibitor, GM6001 will also inhibit other proteases to a certain extent, which explains the marginal reduction in the 34 kDa band intensities. Our finding that Batimastat strongly reduces the formation of the 34 kDa band supports the conclusion that this is a product of MT1-MMP-independent cleavage, likely through ADAM proteases. These results are in line with the current literature discussed in the manuscript on page 8-9, lines 257-259.

And why ADAM15- myc-SNAP was preferentially found in a complex with mCh-MT1-MMP in the presence of TKS4 but not TKS5?

To strengthen the data regarding complex formation between MT1-MMP, TKS4/5 proteins and ADAM15, we have performed additional co-immunoprecipitation experiments using TKS5-GFP and a combination of TKS4- and TKS5-GFP. See answer to point 3 for details.

5) The MDA-MB-231 cell derived CM transferred the invasiveness to non-invasive cells, but whether the CM changed intrinsic MT1-MMP or TKS4/5 expression?

To answer this interesting question, we have performed quantitative RT-PCR analysis of HeLa cells after treatment with CM from MDA-MB-231-TKS5-GFP-mCh-MT1-MMP cells. We could not detect any change in the expression of MT1-MMP or TKS4/5 proteins in the HeLa cells (new Supplementary Fig. 10d).

6) Author should employ more methods and provide more evidence to demonstrate CM changed the invasiveness of recipient cells besides testing the distance in collagen-I. For example, author could conduct trans-well, wound healing etc. and check some genes or proteins involved in cell movement. We thank the reviewer for this excellent suggestion, since the addition of CM could potentially affect the ability of the recipient cells to migrate in general, in an MT1-MMP-independent fashion. We therefore performed experiments where we analysed if the migratory ability of non-invasive cells was changed in the presence of CM from invasive cells. Unlike our finding that addition of CM increases the velocity of cells embedded in Collagen-I, cell migration was not affected when cells were grown on plastic (new Supplementary Fig. 12d). Moreover, the ability to migrate in Collagen-I was strongly inhibited by GM6001, showing that the CM-induced cell migration in Collagen-I depends on protease activity, such as MT1-MMP (new Supplementary Fig. 12a). This supports the conclusion that transfer of invasiveness by CM largely depends on increased ability of the recipient cells to digest ECM, rather than increased ability to migrate.

7) The whole study was completely performed in vitro, despite these current data seems to be solid, the body context will be much more complicated and different compared with in vitro scenario. Author should test TKS4/5 and MP1-MMP mediated intercellular invasiveness transfer happens in vivo as well, of course, this would be incredible hard since the different molecules tracking in vivo is challengeable. However, author should at least determine the impact of TKS4/5 or MP1-MMP depletion on MDA-MB-231 cell invasiveness in vivo and on matrix extracellular matrix degradation. As the reviewer correctly points out, to investigate intercellular transfer of invasiveness in vivo would be extremely challenging. The reviewer suggests that we should at least demonstrate how depletion of MT1-MMP or TKS4/5 proteins in MDA-MB-231 cells would affect invasiveness in vivo, as well as ECM degradation. However, since MT1-MMP and TKS4/5-dependent invasiveness are well established in the literature, with excellent in vivo and in vitro studies, we have chosen to rather discuss this literature in the manuscript on page 2-3 (introduction) and page 13 (end of results section).

In brief: The TKS4/5 proteins were identified as scaffold proteins important for invadopodia formation and their roles in cell invasion and ECM degradation have been established in vitro and in vivo for

several cancer types, such as prostate-, lung-, gastric-, melanoma- and breast cancer¹⁻⁸. Moreover, TKS proteins have been depleted in MDA-MB-231 triple negative breast cancer cells and several other breast cancer cell lines resulting in decreased ability of the cells to degrade ECM and invade in 2D and 3D in vitro models as well as in vivo models^{7,9,10}. Likewise, the invasive properties of the matrix metalloprotease MT1-MMP/MMP14 have been thoroughly studied in vitro and in vivo over three decades, summarized in excellent review articles¹¹⁻¹³. Depletion or inhibition of MT1-MMP in MDA-MB-231 cells has been shown to reduce invasion and ECM degradation in vitro and in vivo^{10,14,15}.

8) To demonstrate MDA-MB-231 derived CM or EVs drive the non-invasive cancer cell to be invasive, author need to inoculate non-invasive cells in mice followed by exogenous CM/EVs injection, then evaluate if invasiveness would be changed.

It would be very interesting, but technically extremely challenging, to study the effect of injected CM on cell invasion in mice. Instead, to strengthen the physiological relevance of our study we have performed co-culture experiments with invasive and non-invasive cells. Importantly, we show that transfer of invasive properties can occur directly between neighbouring cells in culture.

In brief: Non-invasive HeLa cells were seeded on a filter with pores, given the possibility to migrate in a digestive manner into a matrix of polymerized collagen-I, towards a gradient of serum- and HGF-containing medium. Beneath the HeLa cells, in serum free medium, MDA-MB-231 cells were grown, and allowed to condition the medium to support HeLa cell invasion. Importantly, in the presence of MDA-MB-231 cells, the HeLa cells gained invasive properties and were able to migrate through the filter and into the collagen-I matrix. The experimental setup and results are explained in the methods on page 27, and shown in New Fig. 6e.

9) Does this MT1-MMP+TKS4/5 mediated intercellular invasiveness also suite for other types of cancer?

We thank the reviewer for raising this important point. We have expanded our study and now show that TKS4/5- and MT1-MMP-dependent transfer of invasiveness can occur from parental MDA-MB-231 triple negative breast cancer cells and from HT1080 fibrosarcoma cells, to non-invasive HeLa cervix carcinoma cells or MCF7 breast cancer recipient cells (new Figs. 6b,c, 9b,c,f,g, new Supplementary Figs. 9a, 12g,h). Importantly, these experiments also show that transfer of invasiveness does not depend on exogenous overexpression of MT1-MMP or TKS4/5 in the donor cells. Taken together, this supports the notion that the observed TKS4/5- and MT1-MMP-dependent transfer of invasiveness can be a general phenomenon in many cancer types.

10) Do cancer cells from metastatic tumor express higher TKS4/5 compared to one from non-metastatic tumor? Is there correlation between TKS4/5 and survival of patients with cancer?

Yes, indeed, high expression of TKS4/5 and MT1-MMP is associated with cancer progression, metastases and poor survival of patients with several cancer types such as gastric-, head and neck-, prostate-, lung-, melanoma-, breast-, colon-, liver- and ovarian cancers, glioma and leukaemia^{8,15-28}. We have addressed this important knowledge in the manuscript on page 2-3 (introduction) and page 13 (end of results section).

- 1 Seals, D. F. *et al.* The adaptor protein Tks5/Fish is required for podosome formation and function, and for the protease-driven invasion of cancer cells. *Cancer Cell* **7**, 155-165, doi:<https://doi.org/10.1016/j.ccr.2005.01.006> (2005).
- 2 Buschman, M. D. *et al.* The Novel Adaptor Protein Tks4 (SH3PXD2B) Is Required for Functional Podosome Formation. *Molecular Biology of the Cell* **20**, 1302-1311, doi:10.1091/mbc.e08-09-0949 (2009).
- 3 Blouw, B., Seals, D. F., Pass, I., Diaz, B. & Courtneidge, S. A. A role for the podosome/invadopodia scaffold protein Tks5 in tumor growth in vivo. *Eur J Cell Biol* **87**, 555-567, doi:<https://doi.org/10.1016/j.ejcb.2008.02.008> (2008).
- 4 Satoyoshi, R., Aiba, N., Yanagihara, K., Yashiro, M. & Tanaka, M. Tks5 activation in mesothelial cells creates invasion front of peritoneal carcinomatosis. *Oncogene* **34**, 3176-3187, doi:10.1038/onc.2014.246 (2015).
- 5 Iizuka, S., Abdullah, C., Buschman, M. D., Diaz, B. & Courtneidge, S. A. The role of Tks adaptor proteins in invadopodia formation, growth and metastasis of melanoma. *Oncotarget* **7** (2016).
- 6 Leong, Hon S. *et al.* Invadopodia Are Required for Cancer Cell Extravasation and Are a Therapeutic Target for Metastasis. *Cell Reports* **8**, 1558-1570, doi:<https://doi.org/10.1016/j.celrep.2014.07.050> (2014).

- 7 Blouw, B. *et al.* The invadopodia scaffold protein Tks5 is required for the growth of human breast cancer cells in vitro and in vivo. *PLoS One* **10**, e0121003, doi:10.1371/journal.pone.0121003 (2015).
- 8 Burger, K. L. *et al.* Src-dependent Tks5 phosphorylation regulates invadopodia-associated invasion in prostate cancer cells. *Prostate* **74**, 134-148, doi:10.1002/pros.22735 (2014).
- 9 Zagryazhskaya-Masson, A. *et al.* Intersection of TKS5 and FGD1/CDC42 signaling cascades directs the formation of invadopodia. *J Cell Biol* **219**, doi:10.1083/jcb.201910132 (2020).
- 10 Pedersen, N. M. *et al.* Protrudin-mediated ER-endosome contact sites promote MT1-MMP exocytosis and cell invasion. *J Cell Biol* **219**, doi:10.1083/jcb.202003063 (2020).
- 11 Tanaka, N. & Sakamoto, T. MT1-MMP as a Key Regulator of Metastasis. *Cells* **12**, 2187 (2023).
- 12 Knapinska, A. M. & Fields, G. B. The Expanding Role of MT1-MMP in Cancer Progression. *Pharmaceuticals* **12**, 77 (2019).
- 13 Castro-Castro, A. *et al.* Cellular and Molecular Mechanisms of MT1-MMP-Dependent Cancer Cell Invasion. *Annual Review of Cell and Developmental Biology* **32**, 555-576, doi:10.1146/annurev-cellbio-111315-125227 (2016).
- 14 Ling, B. *et al.* A novel immunotherapy targeting MMP-14 limits hypoxia, immune suppression and metastasis in triple-negative breast cancer models. *Oncotarget* **8** (2017).
- 15 Lodillinsky, C. *et al.* p63/MT1-MMP axis is required for in situ to invasive transition in basal-like breast cancer. *Oncogene* **35**, 344-357, doi:10.1038/onc.2015.87 (2016).
- 16 Sugimoto, A. *et al.* The clinicopathologic significance of Tks5 expression of peritoneal mesothelial cells in gastric cancer patients. *PLoS One* **16**, e0253702, doi:10.1371/journal.pone.0253702 (2021).
- 17 Mitre, G. P. *et al.* Key proteins of invadopodia are overexpressed in oral squamous cell carcinoma suggesting an important role of MT1-MMP in the tumoral progression. *Diagn Pathol* **16**, 33, doi:10.1186/s13000-021-01090-7 (2021).
- 18 Ali, A., Soares, A. B., Eymael, D. & Magalhaes, M. Expression of invadopodia markers can identify oral lesions with a high risk of malignant transformation. *J Pathol Clin Res* **7**, 61-74, doi:10.1002/cjp2.182 (2021).
- 19 Li, C. M. *et al.* Differential Tks5 isoform expression contributes to metastatic invasion of lung adenocarcinoma. *Genes Dev* **27**, 1557-1567, doi:10.1101/gad.222745.113 (2013).
- 20 Stylli, S. S., I, S. T., Kaye, A. H. & Lock, P. Prognostic significance of Tks5 expression in gliomas. *J Clin Neurosci* **19**, 436-442, doi:10.1016/j.jocn.2011.11.013 (2012).
- 21 Stylli, S. S. *et al.* Expression of the adaptor protein Tks5 in human cancer: prognostic potential. *Oncol Rep* **32**, 989-1002, doi:10.3892/or.2014.3310 (2014).
- 22 Kui, X. *et al.* Prognostic value of SH3PXD2B (Tks4) in human hepatocellular carcinoma: a combined multi-omics and experimental study. *BMC Med Genomics* **14**, 115, doi:10.1186/s12920-021-00963-6 (2021).
- 23 Makutani, Y. *et al.* Contribution of MMP14-expressing cancer-associated fibroblasts in the tumor immune microenvironment to progression of colorectal cancer. *Front Oncol* **12**, 956270, doi:10.3389/fonc.2022.956270 (2022).
- 24 Wang, X., Meng, Q., Wang, Y. & Gao, Y. Overexpression of MMP14 predicts the poor prognosis in gastric cancer: Meta-analysis and database validation. *Medicine (Baltimore)* **100**, e26545, doi:10.1097/MD.00000000000026545 (2021).
- 25 Kasurinen, A. *et al.* High tissue MMP14 expression predicts worse survival in gastric cancer, particularly with a low PROX1. *Cancer Med* **8**, 6995-7005, doi:10.1002/cam4.2576 (2019).
- 26 Vos, M. C., van der Wurff, A. A. M., van Kuppevelt, T. H. & Massuger, L. The role of MMP-14 in ovarian cancer: a systematic review. *J Ovarian Res* **14**, 101, doi:10.1186/s13048-021-00852-7 (2021).
- 27 Zhang, L. *et al.* Prognostic Significance of Matrix Metalloproteinase 14 in Patients with Cancer: a Systematic Review and Meta-Analysis. *Clin Lab* **66**, doi:10.7754/Clin.Lab.2019.190831 (2020).
- 28 Miguel, A. F. P., Mello, F. W., Melo, G. & Rivero, E. R. C. Association between immunohistochemical expression of matrix metalloproteinases and metastasis in oral squamous cell carcinoma: Systematic review and meta-analysis. *Head Neck* **42**, 569-584, doi:10.1002/hed.26009 (2020).

Reviewer #2 (Remarks to the Author):

This paper from the Raiborg lab describes experiments characterising mechanisms responsible for influencing intracellular trafficking and proteolytic cleavage of the transmembrane matrix metalloprotease, MT1-MMP, and how shedding of MT1-MMP's proteolytic cleavage products can impart invasive characteristics to other cells that do not express MT1-MMP. The authors characterise details of an association formed between MT1-MMP's cytosolic domain and the invadopodia marker proteins TKS4&5, how this depends on TKS4&5's PX domains and how this promotes recruitment of MT1-MMP to the intraluminal vesicles of late endosomes. These protein:protein interactions and their ensuing trafficking to the low pH environment of the late endosome lumen allow ADAM-mediated cleavage of MT1-MMP. The authors then proceed to show that MT1-MMP released from cells – in a manner that is dependent on TKS4&5 – can encourage other cells that do not express MT1-MMP to degrade collagen and become invasive themselves.

In general, the work described in this paper is of great interest, well-executed and convincing and fully support the conclusions drawn by the authors. In particular, the cellular mechanisms influencing how MT1-MMP:TKS4/5 interaction influences MT1-MMP trafficking and how these trafficking events,

in turn, can foster the appropriate conditions to allow ADAM-mediated MT1-MMP cleavage are carefully and convincingly carried-out. The experiments demonstrating that conditioned medium from cells that are competent to execute TKS-dependent trafficking, and thus ADAM-mediated cleavage of MT1-MMP, can encourage HeLa cells to become invasive are also very interesting. However, I feel that more clarification is needed to justify publication of this paper in Nature Comms.

I would highlight my major points 1 and 2 below as being most important. Specifically, that the authors determine:

a) the extent to which the interesting phenomena they describe depend on overexpression of MT1-MMP

We now show that TKS4/5- and MT1-MMP-dependent intercellular transfer of invasiveness can occur between parental cells without exogenous overexpression of MT1-MMP. See answer to major point 1.

and;

b) the role of intraluminal budding and EV production, as opposed to just sorting of MT1-MMP to MVBs/late endosomes, in cleavage of MT1-MMP and intercellular transfer of cancer cell invasiveness.

We now show how intercellular transfer of cancer cell invasiveness depends on functional endosomal trafficking, intraluminal vesicle formation and EV production. See answer to major point 2.

Major points:

1. Most of the experiments are conducted in cells that overexpress an mCherry-tagged MT1-MMP construct. Therefore, the authors should conduct experiments to address the following two questions:

(i) Can TKS-dependent trafficking and cleavage of MT1-MMP be demonstrated for the endogenous transmembrane MMP? This could be determined in cancer cells, in addition to MDA-MB-231s, that endogenously express high levels of MT1-MMP

We agree with the reviewer that it is important to investigate whether the observed TKS4/5-dependent MT1-MMP-cleavage and transfer of intercellular invasiveness can occur in cells that do not exogenously overexpress MT1-MMP. We now show that parental MDA-MB-231 triple negative breast cancer cells and HT1080 fibrosarcoma cells, which both endogenously express high levels of MT1-MMP, are able to transfer invasiveness to the non-invasive, MT1-MMP-negative, HeLa or MCF7 cells in a TKS4/5- and MT1-MMP-dependent manner (new Figs. 6b,c, 9b,c,f,g, new Supplementary Figs. 9a, 12g,h). This implies that TKS4/5-dependent cleavage and shedding of MT1-MMP can occur endogenously, elaborated in answer to major point 1(ii), below.

(ii) Does the inclusion of the mCherry tag in the juxtamembrane ectodomain influence MT1-MMP's TKS-dependent trafficking and cleavage?

Since tagging of MT1-MMP on its N- or C-terminal ends might affect its trafficking and function, internal tagging of MT1-MMP is considered less disturbing and is widely used in MT1-MMP research. Nevertheless, the reviewer is correct, that the inclusion of the mCherry tag in the internal juxtamembrane linker region of MT1-MMP could potentially influence the TKS4/5-dependent cleavage and shedding of the ectodomain.

To test whether the internal mCherry tag in MT1-MMP affected TKS4/5-dependent shedding, we generated an MDA-MB-231 cell line stably expressing untagged MT1-MMP. Importantly, a prominent 50-52 kDa anti-MT1-MMP positive product was detected in the conditioned medium from these cells, which was significantly reduced upon TKS4/5 depletion (new Fig. 4g, new Supplementary Fig. 7f, g). A weak TKS4/5-dependent 50-52 kDa product was also detected in conditioned medium from parental

MDA-MB-231 cells (new Supplementary Fig. 7h). Moreover, Zymography measurements of the conditioned medium showed that the 50-52 kDa products possess enzymatic activity, which was significantly reduced upon TKS4/5 depletion in parental MDA-MB-231 cells as well as in MDA-MB-231 cells stably expressing untagged MT1-MMP (new Fig. 9i, new Supplementary Fig. 12g).

Taken together, we conclude that TKS4/5-dependent cleavage can occur in untagged or endogenous MT1-MMP, that the cleavage product is enzymatically active, and that intercellular transfer of invasiveness occurs from highly invasive cancer cells without exogenous overexpression of MT1-MMP. These findings validate the use of the mCherry-tagged MT1-MMP in this study.

2. The authors show that MT1-MMP and some of its cleavage fragments are released from cells both in association with EVs and in solution. They also demonstrate in Fig. 6e that both EV-associated and soluble MT1-MMP (fragments) can transfer invasiveness to HeLa cells. In these regards, can the authors address the following questions:

(i) To what extent are ILV-budding and EV production/release actually required for TKS-dependent cleavage of MT1-MMP and release of MT1-MMP (fragments) capable of intercellular transfer of invasiveness?

We thank the reviewer for raising this important point. We have performed new experiments that address the role of ILV-budding and EV production/release for the shedding of MT1-MMP fragments and transfer of invasiveness. The new data are presented in a new Fig. 8 and new Supplementary Fig. 11.

In conclusion, we observed that MT1-MMP ectodomain-release was unaffected in the absence of ILV production in cells co-depleted for the ESCRT-recruiter HRS and CD63, whereas EV release was reduced, as expected. These cells were still capable of transferring invasiveness, although to a lesser extent than control treated cells. This indicates that the shed ectodomain is to some extent capable of intercellular transfer of invasiveness in the absence ILV formation and EV release, consistent with our previous fractionation experiment (old Fig.6e, new Fig. 8a).

Moreover, when we inhibited both EV- and ectodomain- release by depleting RAB7, important for endosome maturation and trafficking, the cells were virtually incapable of transferring invasiveness. This implies that most of the observed intercellular transfer of invasiveness indeed depends on endosomal function and trafficking.

Taken together, our results show that both EVs and the shed MT1-MMP ectodomain can mediate transfer of invasiveness, that they are more potent together, and that the presence of both in the conditioned medium depends on functional late endosomes.

For more details, see answer to major point 2(ii and iii), below.

(ii) How does interfering with intraluminal budding, for instance by KD/KO of CD63, Tsg101 and/or ESCRTs affect MT1-MMP cleavage/release? The work would also be strengthened if the authors included markers such as CD63, Alix or ESCRT-related proteins to show endocytic origin in their EV preps (note: the CD9 antibody they are using has been discontinued due to low specificity and the blots are very faint for CD9 in WCL, for instance in figure 3c).

As suggested, we interfered with ILV budding by co-depleting cells for HRS and CD63, which should impair ESCRT- and CD63-dependent ILV formation, respectively. As expected, this treatment reduced the release of extracellular vesicles into the conditioned medium, as detected by the use of anti-Syntenin-1 in the conditioned medium (new Fig. 8c, new Supplementary Fig. 11a). Syntenin-1 interacts directly with the ESCRT protein Alix, involved in ESCRT-dependent ILV formation, and should serve as a reliable marker for endosome-derived EVs.

Interestingly, the MT1-MMP ectodomain was still released from cells with reduced ILV/EV formation. Since MT1-MMP and TKS5 localized to the limiting membrane of endosomes in HRS/CD63 co-depleted cells (new Fig. 8b), our data indicate that TKS4/5-dependent cleavage can occur in endosomes in the absence of ILV formation, releasing the ectodomain into the medium upon endosome-fusion with the plasma membrane.

We thank the reviewer for enlightening us regarding the use of the CD9 antibody (ab92726 from Abcam). We have now re-probed the blot in Fig. 3c, using the anti-Syntenin-1 antibody, and a different CD9 (ab263019 from Abcam) antibody, which is KO validated by the manufacturer. In addition, we now include CD63 as a marker for EVs in the new Supplementary Fig. 12f. Further, we characterize EVs by electron microscopy, and show that they label positive for CD63 (see answer to Major point 4 for details). These experiments confirm the endocytic origin of the EVs.

(iii) What is the effect of knockdown of Rab27s, or other ways to oppose LE/MVB docking/fusion (such as VAMP7 knockdown) on release of MT1-MMP fragments and intercellular transfer of invasiveness?

To prevent fusion of late multivesicular endosomes with the plasma membrane, we chose to deplete cells for Protrudin or RAB7, since these tools are efficient and well established in our lab (new Supplementary Fig. 11). Protrudin is a protein required for the translocation of late endosomes to the plasma membrane. The small GTPase RAB7 is required for late endosome function and transport. In Protrudin- or RAB7-depleted cells, TKS5 and MT1-MMP accumulated in enlarged endosomes in the perinuclear area (new Fig. 8b). Accordingly, the conditioned medium contained reduced amounts of the EV marker Syntenin-1 and less of the shed ectodomain (new Fig. 8c,d). Moreover, RAB7 depleted cells were not capable of transferring invasive properties to non-invasive cells (new Fig. 8e). Thus, intercellular transfer of invasiveness requires functional endosomal transport to (and subsequent fusion with) the plasma membrane.

We cannot rule out that a fraction of the shed ectodomain and EVs comes from the plasma membrane. However, since RAB7-depletion inhibited EV/ectodomain-release and prevented transfer of invasiveness, this indicates that the majority of the TKS5-dependent cleavage of MT1-MMP occurs in acidic endosomes. This is in line with our finding that MT1-MMP recruits TKS4/5 to endosomes.

3. The authors have shown very nicely that non-MT1-MMP1-expressing recipient cells (HeLas) are required for released MT1-MMP fragments to promote invasion. But what happens to soluble and EV-associated MT1-MMP fragments when they encounter the HeLa cells? Do they become associated with domains on the HeLa cell surface? Are they internalised by HeLas and, if so, is this required for the HeLas to enact collagen degradation/invasion?

These are indeed interesting questions, which we partially addressed in the previous version, and have now characterized further in the revised version of the manuscript (new Fig. 7).

By immunofluorescence (IF) imaging, we detected the full-length (EV-derived) mCherry-MT1-MMP on the outside of recipient cells, and even inside endosomes of the recipient cells (old Fig. 5c,d, new Fig. 7c,e, new supplementary Fig. 10). The detection of the MT1-MMP fragment in recipient cells by IF, however, unfortunately failed due to low sensitivity of the anti-MT1-MMP antibody. Since N-terminal tagging of MT1-MMP should be avoided to ensure proper localization and processing, we have very limited tools to characterize the localization of the MT1-MMP ectodomain by IF. Instead, as the MT1-MMP antibody works well on Western blotting (verified by knock down experiments), we could show that the MT1-MMP fragments from the conditioned medium indeed associate with recipient HeLa cells (old Fig. 5e, new Fig. 7f).

Fractionation experiments showed that the MT1-MMP fragments exist in the soluble- and EV-fraction (old Fig. 6e, new Fig. 8a). Since we were not able to follow the fate of the soluble MT1-MMP fragments in recipient cells, we continued to characterize the EV-associated pool of MT1-MMP using the mCherry-tag. We now find that EV-associated MT1-MMP partly co-localizes with the focal adhesion marker Zyxin at the surface of recipient HeLa cells (new Fig. 7d). Moreover, the pattern of degraded gelatin partly overlaps with actin fibres of the HeLa cells (new Fig. 7b). Taken together, we find that mCherry-MT1-MMP positive EVs dock on focal adhesions at the surface of recipient cells, where they contribute to ECM degradation, in a pattern resembling actin fibres. These areas of tight contact between cells and substrate likely enable continuous and direct vicinity between cell associated metalloprotease molecules and their substrate, allowing ECM digestion.

Interestingly, HeLa-associated gelatin degradation required living HeLa cells. Formaldehyde fixation before treatment with conditioned medium greatly reduced the ability of the recipient cells to degrade gelatin, leaving a weak and diffuse cell-associated trace, unlike the focal adhesion-like pattern observed in living cells (new Fig. 7a).

Is the surface localization sufficient to perform ECM degradation, or does it depend on the internalization of mCherry-MT1-MMP positive EVs or MT1-MMP fragments by the recipient cells? We addressed this very interesting question, raised by the reviewer, by impairing endocytosis in HeLa cells and assessed their ability to degrade gelatin in the presence of conditioned medium from invasive cells (new Fig. 7g,h, Supplementary Fig. 10e-g). Inhibition of Clathrin mediated endocytosis did not affect HeLa-associated gelatin degradation. When global endocytic uptake was perturbed by hypertonic treatment, the HeLa-associated degradation was partly reduced. Thus, intercellular transfer of invasiveness can happen when uptake is inhibited, but is more efficient when the recipient cells are not stressed and able to perform endocytosis. A possible explanation could be that the degradation is improved by transcytosis of MT1-MMP. Alternatively, uptake of EVs might trigger a transcriptional response in the HeLa recipient cells.

Quantitative RT-PCR analysis of HeLa cells treated with conditioned medium from MDA-MB-231-TKS5-GFP-mCh-MT1-MMP cells however, did not reveal any change in the expression of MT1-MMP or TKS4/5 (new Supplementary Fig. 10d).

4. What is the topology of MT1-MMP and TKSs in EVs? Can the authors determine this using immuno-gold EM?

Immuno-gold labelling of sectioned multivesicular endosomes showed that the intraluminal vesicles (ILVs) were positive for mCherry-MT1-MMP and TKS5-GFP (old and new Fig. 3b). TKS5-GFP was detected in the lumen of the ILVs, consistent with its interaction with the cytosolic tail of MT1-MMP. This topology should be identical in EVs whether they are derived from ILVs or from the plasma membrane.

In the revised version of the manuscript, we have characterized the EVs by electron microscopy, and find that the majority have a diameter between 40 and 100 nm, consistent with the size of ILVs. Moreover, immuno-gold labelling of the EVs shows that the 40-80 nm EVs had the highest density of CD63 labelling (new Fig. 3d). The EVs analysed by EM are not permeabilized or sectioned. Thus, luminal proteins will not be accessible for immuno-gold labelling. Whereas the transmembrane mCherry-MT1-MMP was detected at the surface of the EVs, the EVs stained negatively for TKS5-GFP (new Fig. 3d). This is consistent with a localization of TKS5-GFP in the lumen of the EVs. This is supported by the previous EM analysis on sections, detecting TKS5-GFP in the lumen of ILVs, and the presence of endogenous TKS4 and TKS5 in the EV fraction by Western blotting.

5. I'm not sure I agree that Marimastat is more potent than GM6001 in generating the 34kDa fragment of MT1-MMP. From the blot in Fig. 4g it looks very similar to GM6001.

We have modified this statement on page 8, lines 256-257.

6. I don't think that the data presented indicate that the TKs PX domain alone is sufficient for MT1-MMP-dependent endosomal localisation. From Fig. 2b it looks as if TKs 1-139 is recruited to endosomes in the presence of MT1-MMP whether it possesses the (PX-binding) cytotail...so this looks non-specific to me. Better to say that the PX domain is necessary, but not necessarily sufficient, for endosomal localisation of TKS5.

We agree with the reviewer and have modified this statement on page 5, lines 139.

Minor comments:

1. It would be helpful if the authors could indicate, in addition to the constructs used for stable expression in Fig. 2a, the siRNA targeting the endogenous MT1-MMP (as stated in the main text and respective figure legend).

We have now indicated this in new Fig. 1e (old Fig. 2a).

2. In figure 6F the authors should justify the use of the term "exosomes" instead of EVs when explaining the schematic model of this figure.

Since it is difficult to distinguish between exosomes or microvesicles, we use the term EVs throughout the manuscript. As our data point to a role of endosomes in TKS-dependent MT1-MMP shedding and transfer of invasiveness, we chose to use the expression "exosomes" in the explanation to the model figure.

Through the revision process, based on constructive feedback from the reviewers, we have strengthened the data regarding the endosomal contribution to our observed mechanism. Characterization of the EVs by EM shows that the majority of the EVs have a diameter of 40-100 nm, with a high density of CD63 labelling, consistent with an endosomal origin (new Fig. 3d). This notion is further strengthened by functional experiments where we abrogate ILV formation and/or exosome secretion and observe a reduced transfer of invasiveness (new Fig. 8). Thus, we suggest to keep the expression "exosomes" in the explanation to the model figure.

REVIEWERS' COMMENTS

Reviewer #1 (Remarks to the Author):

The author has effectively addressed all my concerns by furnishing comprehensive data and providing thorough responses to my concerns. Nevertheless, to enhance the replicability of Western Blot (WB) data, I recommend incorporating details on the loaded protein amount in the gel alongside an unaltered image.

Reviewer #2 (Remarks to the Author):

The authors have satisfactorily addressed all of my concerns, and I would like to offer my congratulations to the authors on the execution of such a fine study.